

# Time Series Analysis of Ground-Based Microwave Measurements at K- and V-Bands to Detect Temporal Changes in Water Vapor and Temperature Profiles

Sibananda Panda[1], Swaroop Sahoo[2], G. Pandithurai[3]

[1]School of Electronics Engineering, KIIT University, Odisha, India
[2]Department of Electronics and Communication Engineering, Amrita University, Tamil Nadu, India
[3]Indian Institute of Tropical Meteorology, Pune, India

*Correspondence to*: Swaroop Sahoo (swaroop.sahoo769@gmail.com)

**Abstract.** Ground-based microwave measurements performed at water vapor and oxygen absorption line frequencies are widely used for remote sensing of tropospheric water vapor density and temperature profiles, respectively. This paper focuses on using time series of independent frequency measurements at K- and V-bands along with statistically significant short background data sets to sense temporal variations and gradients in water vapor and temperature profiles. To study this capability, Indian Institute of Tropical Meteorology (IITM) had deployed a microwave radiometer from Radiometric Corporation at Mahabubnagar, Hyderabad during August 2011. In this study, time series of water vapor and temperature were retrieved using Bayesian optimal estimation method which uses Levenberg-Marquardt optimization technique. The temperature profile has been estimated using optimized background information covariance matrix for the first time to improve the accuracy of the retrieved profiles. Estimated water vapor and temperature profiles are compared with profiles of same parameters taken from the reanalysis data updated by National Oceanic and Atmospheric Administration (NOAA), Earth System Research Laboratory. RMS errors are evaluated for the water vapor and temperature profiles for a month. It is found that water vapor and temperature profiles can be estimated with an acceptable accuracy by using a background information data set compiled over a period of one month.

*Keywords*-*microwave radiometer, humidity, temperature and Bayesian optimal estimation*

## 1. INTRODUCTION

Water vapor along with temperature affects various atmospheric processes, particularly cloud formation, initiation of convective storms (Trenberth et. al., 2005) and tropical cyclones (Needs, 2009) (Ali, 2009). Therefore, accurate information about their spatial and temperal distribution as well as variation in the lower troposphere is essential for initialization of numerical weather prediction (NWP) models which in turn improve the forecast of various weather events (NRC, 2009).

Various instruments are used to measure water vapor and temperature profiles in the lower troposphere i.e., radiosondes, Raman Lidar and microwave radiometer. Radiosondes are by far the main source of water vapor and temperature profiles information for numerical weather prediction. The measured profiles have a vertical resolution of approximately 10 m in the lowest 3 km of troposphere but are launched once or twice a day at most of the site in the world (Wang et. al., 2008).



Therefore, they cannot be used to detect the temporal variations and gradients in the atmospheric humidity and temperature profiles at regular intervals of time. Raman lidars (Goldsmith et. al., 1998) are also used during clear sky conditions for sensing humidity profiles with a vertical resolution comparable to that of a radiosonde from ground to an altitude of 3 km but are quite expensive to deploy in a dense network so as to be able to provide information on spatial distribution and variation

on water vapor and temperature. In addition to these instruments, microwave radiometers, both ground-based and airborne, operating in the 20-60 and 166-190 GHz ranges are used for retrieving water vapor, temperature and liquid water profiles. Ground-based microwave radiometers have been designed, fabricated and used to sense water vapor and temperature profiles from ground to 10 km altitude (Iturbide-Sanchez et. al., 2007) (Solheim, et al., 1998). Satellite based instruments like AMSU-A and –B onboard NOAA-15 (Susskind et.al., 2011) (Rosenkranz, 2001) as well as SAPHIR-MADRAS onboard the

Megha-Tropiques (Brogniez, et. al., 2013) have been used to retrieve humdity and temperature profiles in addition to a range of other parameters. The AMSU-A and –B channels operate close to the 22.235, 60, 183 GHz absorption lines as well as the 89 GHz window frequency. In addidtion to these instruments a constellation of minisatellites known as FLORAD operates at frequencies close to the 89, 118 and 183 GHz to estimate water vapor, temperature, cloud liquid content and precipitation rate (Marzano, et al., 2009).

Both the ground-based and airborne microwave radiometers have a fine temporal resolution ranging from a few millisecond to a few minutes depending on the integration time of the measurement channel as well as the mode of deployment i.e., ground-based or airborne. However, radiometers have a variable vertical resolution and accuracy depending on the thermodynamic property being retrieved.

Humidity and temperature profiles have been retrieved from microwave radiometer measurements by Westwater

(Westwater, 1993) while Scheve (Scheve et al., 1999) used the minimum variance estimation technique. Sahoo (Sahoo et al., 2015b) used the Bayesian optimal estimation technique while focusing on sensing the gradients and temporal changes associated with water vapor profiles retrieved by using K-band radiometer measurements and an optimized background datasets.

The novel feature of this work is the estimation of water vapor density and temperature profiles within certain limits of

accuracy while detecting temporal variations and gradients in the profiles. The profiles are estimated by inverting K- and V-band measurements using the Bayesian algorithm along with a background data set compiled over a period of one month. The background dataset is specific to the time period of radiometer measurement and conforms to the weather conditions during that period. This method results in a significant improvement of accuracy over the normal method of using a large dataset collected for a long period of time as background dataset. However, the improved results are specific for a location

and the method needs be adapted for a particular region.

The Bayesian optimally estimated profiles are then compared with profiles estimated using the neural network method as well as profiles taken from the reanalysis data from Earth System Research Laboratory, National Oceanic and Atmospheric Administration (NOAA). Here, the reanalysis data is considered as truth and the error in this study is the difference between



the radiometer retrieved profiles (using both neural network and Bayesian optimal estimation) and those from the reanalysis data.

## 2. INSTRUMENTS DEPLOYMENT

Indian Institute of Tropical Meteorology (IITM) had deployed a microwave radiometer in Mahabubnagar (16° 44′ N, 77° 59′ E), Hyderabad for the whole month of August, 2011. This microwave radiometer, MP-3000A has been procured from Radiometric Corporation, Boulder, Colorado, USA (Liljegren, 2002). The radiometer operates at 8 frequencies in the range 22-30 GHz and 14 frequencies from 51.0 GHz to 58.0 GHz at V-band, at elevation angle range of $15^o$, $90^o$ and $165^o$. This instrument also has a single channel infrared radiometer in addition to surface pressure, humidity and temperature sensors. The multichannel microwave radiometer is calibrated by using noise diode injection to remove the system gain fluctuations. Two sided tipping curve calibration method has been used to determine the brightness temperatures from the measured voltages.

Radiometer measurements were performed throughout the day and night under varying atmospheric conditions which included clear and cloudy sky. The time series of brightness temperatures for 22.23, 25.0, 51.243 and 53.36 GHz are shown in Figure 1. It can be observed that antenna brightness temperatures at 22.23 GHz are comparatively higher than those at 25 GHz. This is because 22.23 GHz is the water vapor resonance frequency and is more sensitive to water vapor in the atmosphere than 25 GHz which is significantly far away from the water vapor resonance frequency. Similarly, measurements at 53.36 GHz are higher than those at 51.243 GHz because of the proximity of 53.36 GHz to the oxygen complex. Thus, measurement frequencies are sensitive to water vapor and temperature to a varying extent. This can also be confirmed by analyzing the weighting functions corresponding to water vapor and temperature frequencies shown in Figure 2. Figure 2 (a) shows that 22.234 and 24.034 GHz measurements are sensitive to water vapor changes 5 km above ground level while 25 and 26.343 GHz measurements are more sensitive to changes in the lowest one km of troposphere. However, the temperature measurement frequencies shown in Figure 2 (b) are most sensitive to temperature variations from 0 to 4 km altitude.

To complement the radiometer measurements, Vaisala RS92-SGP radiosondes were launched everyday at 12 UTC. These radiosondes were launched from the radiometer deployment location to provide vertical profiles (temporal resolution of two seconds) of relative humidity, temperature, dew point temperature, pressure, and wind. These radiosondes data have been used as the source of apriori information as well as the source of background dataset during this study and analysis.

## 3. THEORETICAL BACKGROUND

### 3.1 Impact of Background Dataset on Retrieval

As already studied and determined by Scheve (Scheve & Swift, 1999), Hewison (Hewison, 2007), Solheim (Solheim, et al., 1998) and Sahoo (Sahoo et. al., 2015a) the number of measurement frequencies which provide altitude related information about water vapor and temperature are limited by the information content or degrees of freedom of the measurements. Thus,



use of these measurements in various inversion methods to retrieve the thermodynamic properties at more number of altitudes than the information content limit is an ill-posed as well as a non-linear problem. This results in an infinite number of possible estimated profiles within the limits of the observation error for various optimization and statistical methods. Because of the non-linearity of humidity and temperature retrieval, statistical regression methods like neural network

estimation are liable to increase in errors if used beyond the range of training data set. Similarly, optimization methods also have problems related to slow convergence which further worsen with cloudy conditions (Rodgers, 2000).

Since, the retrieval of water vapor and temperature profiles is a non-linear problem; a variation of the iterative Gauss-Newton method is required for the estimation process. This retrieval technique uses a-priori humidity and temperature information as a constraint to determine an unique solution to the inverse problem. In addition to the a-priori information,

water vapor density and temperature background information statistics is also introduced as the inversion constraint. These statistics provide variability information associated with the atmospheric humidity and temperature profiles as well as the inter layer correlation during a particular time period.

The background statistics is in the form of background information covariance matrix. This matrix is calculated from a dataset of humidity and temperature profiles measurements over a particular time period. This resultant covariance matrix is

very important for the performance of the retrieval algorithm in terms of accuracy and ability to sense temporal changes. Covariance matrix computed from a small dataset which is statistically significant can be used to retrieve water vapor and temperature profiles so as to sense temporal variations and gradients in the water vapor and temperature profile. Figure 3 (a) and (b) show the background information covariance matrix calculated using water vapor density and temperature profiles measured over a period of one month, respectively. It can be observed that most of the water vapor variability information is

between 20-40 layers which correspond to the altitude range of 2 to 4 km. On the other hand, Figure 3 (b) shows that the temperature variability information is primarily below an altitude of one km and also in the range of 2-4.2 km. In contrast to these results, when background information covariance matrix is computed from a large dataset, the accuracy of the retrieved profile is improved because of higher order statistical information However, important weather events or temporally varying conditions are overshadowed because the covariance matrix takes into the consideration the overall variability information 5

while reducing the weight of certain weather conditions which correspond to a particular season (Sahoo et.al., 2015b).

However, the goal of this study is to retrieve water vapor and temperature profiles with improved accuracy while using a background dataset measured over a period of one month so as to detect the temporal changes and gradients in the lowest 8 km of the profiles.

## 3.2 Retrieval Technique

### 3.2.1    Bayesian Optimal Estimation

In the Bayesian optimal estimation, multiple K- and V-band microwave frequency measurements are used to retrieve profiles of humidity and temperature. This is because various measurement frequencies have varying sensitivity to water vapor and temperature at various altitudes. These multiple measurements combined with the a-priori information and a background

information covariance matrix can be used for retrieving water vapor and temperature profiles while sensing the associated temporal variations and gradients.

The Bayesian optimal estimation uses the Levenberg-Marquardt (LM) optimization method (Rodgers, 2000) and has also been used in this paper as shown in Eqs. (1), (2) and (3).

$$\bar{x}_{i+1} = \bar{x}_i + \left((1+\gamma)\bar{\bar{S}}_a^{-1} + \bar{\bar{K}}_i^T \bar{\bar{S}}_\epsilon^{-1} \bar{\bar{K}}_i\right)^{-1} \left(\bar{\bar{K}}_i^T \bar{\bar{S}}_\epsilon^{-1}[\bar{T}_B' - \bar{T}_B(\bar{x}_i)] - \bar{\bar{S}}_a^{-1}[\bar{x}_i - \bar{x}_a]\right)$$ (1)

5 where $i$ is the iteration index, $\bar{\bar{K}}_i$ is the kernel or weighting function matrix and determines the sensitivity of the measurements at various frequencies to changes in the parameter of interest at various altitudes, $\bar{x}_i$ is the initialization profile which could either be water vapor density or temperature profiles, $\bar{T}_B'$ is the measured brightness temperature vector at water vapor density or temperature frequencies, $\bar{x}_a$ is the background profile and is same as the initialization profile in this case because a small dataset is used as background dataset. $\bar{T}_B(\bar{x}_i)$ is the radiative transfer model simulated brightness 10 temperature using the absorption coefficients calculated from a Rosenkranz model (Rosenkranz, 1993) (Rosenkranz, 1998). $\bar{\bar{S}}_\epsilon$ is the observation error covariance matrix and contains the measurement error information. The observation error covariance matrix takes into consideration the radiometric measurement noise, representativeness error and radiative transfer model errors. The diagonal elements of the observation error covariance matrix are calculated as the radiometric resolution of each measurement channel while the off-diagonal elements are approximately zero because the measurements are 15 independent of each other in this analysis. The diagonal elements of the matrix are approximately 0.25 K at all the frequencies (Liljegren, 2002) of interest. $\bar{\bar{S}}_a$ is the background covariance matrix which is computed using information from 50 radiosonde profiles launched over a period of one month. $\gamma$ is the LM factor and its value of $\gamma$ is updated at each iteration based on value of $J(x)$ from (2). If for a particular iteration, the value of $J(x)$ increases, then the iteration is discarded and the value of $\gamma$ is increased 10 fold and the iteration is repeated. If value of $J(x)$ decreases, then the iteration is valid and the 20 value of $\gamma$ is reduced by a factor of 2 for the next iteration (Hewison, 2007). Two values of $\gamma$ have considered been considered for starting of the iteration. For $\gamma = 1$, the iteration moves towards a local minima while in case of $\gamma = \infty$ the iteration immediately moves towards the global minima which gives a solution which does not converge. Therefore, an initial value of $\gamma$ is assumed to be one.

LM technique is an iterative process where the $\gamma$ value is updated at each iteration so that the cost function represented by $J$ 25 in Eq. (2) reaches a minimum value

$$J(x) = [\bar{x} - \bar{x}^b]^T \bar{\bar{S}}_a^{-1}[\bar{x} - \bar{x}^b] + [\bar{T}_B(\bar{x}_i) - \bar{T}_B']^T \bar{\bar{S}}_\epsilon^{-1}[\bar{T}_B(\bar{x}_i) - \bar{T}_B']$$ (2)

where $\bar{x}^b$ and $\bar{x}$ are the initialization profiles (either water vapor or temperature) and output profiles (either water vapor or temperature) for each iteration, respectively. The final water vapor or temperature output profiles are determined by the convergence criterion given by Eq. (3)

$$[\bar{T}_B(\bar{x}_{i+1}) - \bar{T}_B(\bar{x}_i)]^T \bar{\bar{S}}_{\delta y}^{-1}[\bar{T}_B(\bar{x}_{i+1}) - \bar{T}_B(\bar{x}_i)] \ll m$$ (3)

where $m$ is 5 and 7 for water vapor and temperature profile retrieval and $\bar{\bar{S}}_{\delta y}$ is the covariance between $\bar{T}'_B$ and $\bar{T}_B(\bar{x}_i)$. The iteration stops when (3) reaches a value of 0.05 and the output profiles are consistent with atmospheric conditions.

### 3.2.2 Neural Network Estimation

5   Estimation of water vapor and temperature profiles from microwave radiometer brightness temperatures using neural network method (Solheim, et al., 1998) requires training of the algorithm using 4 to 5 years data set of radiosonde measured humidity and temperature profiles as well as profiles of liquid water taken over a period of 5 years. These profiles are then used to simulate brightness temperatures using a radiative transfer model which are then used to train the NN. However, sufficient radiosonde profiles were not available for Mahabubnagar, so a slightly different approach was used in this study for neural network estimation of profiles. Radiosonde profiles were still used as training dataset but these were taken from areas which had similar weather conditions as Mahabubnagar, Hyderabad as well as same altitude and latitude (but different longitude). However, two sites at the same altitude and longitude may have significantly different weather depending on the general conformation of the mountains in the area, the marine currents as well as the advection processes. This could lead to biases in the training of the radiometer algorithm which in turn would increase the error of the retrieved profile. The profiles of water vapor and temperature are retrieved using proprietary NN zenith software developed by Radiometrics Corporation. NN zenith estimation of temperature, water vapor density, relative humidity, and liquid water content profiles are performed simultaneously from the radiometer measurements plus the IR channel.

### 4.   RETRIEVAL OF ATMOSPHERIC PROFILES

20   The Bayesian optimal estimation and NN zenith method are applied to the zenith microwave measurements to estimate water vapor density and temperature profiles from ground level to 8 km altitude of the troposphere for various days and times. The profile estimation method requires an initialization profile which is taken from radiosondes launched every day at 12:00 UTC. The initialization profiles from radiosondes are vertically averaged to correspond to 100 m layer thickness of retrieval. In addition to the measurements and initialization profile, background information covariance matrix is also required which is calculated from the dataset of radiosonde profiles launched during the experiment. The retrieved profiles are compared with NOAA reanalysis data. The reanalysis profiles have water vapor and temperature samples at varying pressure levels. These profiles are made uniform by interpolation so as to have samples at every 100 m interval from ground to 8 km above ground level.

### 4.1 Water Vapor Profiles

Water vapor profiles estimated using the Bayesian optimal estimation and neural network method are shown in Figure 4, along with the reanalysis data from NOAA. It can be observed that Bayesian optimal estimation performs better than the NN zenith in estimating the water vapor profile on all the days considered. The Bayesian optimal estimation is able to detect the variation in the profiles which are smoothed by the NOAA data because of the coarse vertical resolution. Figure 4 (a) shows





that both the Bayesian and NN zenith estimated water vapor profile have similar performance on 7-August-2011 when the errors are in the range of 1.5-2.5 g/m$^3$ from ground to 3 km above ground level. However, for 16-August-2011 the Bayesian and NN zenith retrieval errors are in the ranges of 0-1.5 and 0-3 g/m$^3$, respectively, as shown in Figure 4 (b). The Bayesian retrieved profiles show significantly improved performance on 25-August-2011 and 26-August-2011 and have an error less than 1.5 g/m$^3$, in the lowest 2 km of the troposphere which is better than the error associated with the NN zenith estimated profiles. It can also be observed from Figure 4, that at the altitude range of 3-7 km, Bayesian optimal estimation method has an error less than 0.8 g/m$^3$ except on 25-August-2011. Thus the NN zenith retrieval has a slight negative bias in the lowest 2 km of the troposphere.

In addition to the higher accuracy, Bayesian optimal estimation has been able to detect the gradient in the lowest 4 km of troposphere on 7, 16 and 25-August-2016 which is not observed in the reanalysis data. This detection of gradients in the water vapor density profiles is significant because water vapor is highly variable in the lowest 3 to 4 km of the troposphere which greatly affects the evolution of the weather changes. The statistical analysis for profiles retrieved by Bayesian optimal estimation and NNzenith is discussed in Section 4C.

## 4.2 Temperature Profiles

Temperature profiles were estimated using both the estimation methods (Bayesian and NN zenith method) and have been shown in Figure 5. The NN zenith estimated method consistently shows a negative bias when compared with the reanalysis data at all altitudes. However, the negative bias is more significant above 3 km of the troposphere where the NN zenith error is higher than 6 K. On the other hand, the Bayesian method outperforms the NN zenith estimated profile on all the days considered here. For altitudes below 3 km, the Bayesian optimal estimation has an error range of 0-1.5 K and for altitudes above 4 km, the Bayesian method error is less than 3 K for all the cases considered in Figure 5. It can be noted that some of the fine changes and gradients in the temperature profile in the altitude range of 0-3 km are sensed by the Bayesian estimated temperature profile. The ability to sense gradients and temporal changes in the temperature profile are because of the background covariance matrix which has been computed using a dataset compiled over one month during the measurement time. This dataset is correlated to the radiometer measurements because the radiosondes data have been taken during the measurements. Considering that the reanalysis profiles are true temperature profiles, the optimal estimation has better retrieval accuracy than that of NN zenith in the whole troposphere. The statistical analysis for Bayesian optimal estimation and NN zenith is discussed in Section 4C.

## 4.3 Error Analysis

To analyze the performance of both the retrieval techniques, the retrieval errors are calculated as the difference between the estimated (using either the NN zenith or the Bayesian optimal estimation) and the reanalysis profiles. The range of errors associated with the water vapor profiles estimated using NN zenith and Bayesian optimal are shown in Figure 6 (a) and (b), respectively. Figure 6 (a) shows that the errors in the lowest 2 km of the troposphere are in the range of -4.5 to 4 g/m$^3$ and as





the altitude increases the error range decreases and reduces to -2 to 2 g/m$^3$ at 4 km above ground level. However, the range of errors associated with the Bayesian optimal estimation are less than -1 to 1 g/m$^3$ at all altitudes for most of the cases as shown in Figure 6 (b). Thus, the error associated with the Bayesian optimal estimation is significantly less than that of the NN zenith algorithm particularly for water vapor profile retrieval. It can also be observed that the errors in most of the cases are less than zero for both the retrieval methods. This is because of the estimated profile being less than the reanalysis profile in most of the cases (thus the negative bias).

Figure 7 (a) and (b) show the range of errors associated with the temperature profiles estimated using the NN zenith and Bayesian optimal techniques, respectively. The error associated with the neural network profile is in the range of -3 to 5 K in the lowest 1 km of the troposphere and then the range changes to -4 to -8 K at 4 km above ground level. It is clear that the NN zenith retrieval underestimates the value of the temperature profile. The error associated with the Bayesian optimal estimated profile is shown in Figure 7 (b) and is in the range of -1 to 0 K except in the case of a few profiles. As in the case of water vapor profile, the errors associated with temperature profiles by Bayesian optimal estimation are significantly less than NNzenith estimated profiles. The Bayesian optimally estimated retrievals using radiometer observations compare well with the reanalysis data in the lowest 2 km of troposphere because of the retrieval being constrained by surface measurements provided by the radiometer.

To determine the deviation of the retrieved profiles from the reanalysis profile, root mean squared (RMS) errors are calculated for both the methods. RMS errors are calculated by comparing radiometer retrieved humidity and temperature profiles (retrieved using both Bayesian optimal estimation and NNzenith method) with the reanalysis data (which is used as truth in this case) and are shown in Figure 8. Figure 8 (a) shows that RMS error associated with Bayesian optimal estimated water vapor profile varies from 0.2 to 0.4 g/m$^3$ for the lowest 4 km of the troposphere and is less than 0.2 g/m$^3$ above 5 km altitude. On the other hand, RMS error for NN zenith retrieved profile is in the range of 1-2.5 g/m$^3$ in the lowest 2 km and is less than 1 g/m$^3$ above 4 km. Thus, the RMS error for water vapor profile retrieved using Bayesian optimal estimation is better than NN zenith.

It can be observed from Figure 8 (b) that the RMS error for Bayesian optimal estimated temperature profiles is less than 0.6 K at any altitude from 0 to 8 km above ground level. However, the NN zenith retrieval error range is 1-2 K for lowest 2 km and then increases consistently above 2 km. The maximum error is approximately 7.5 K at 8 km above ground level. Thus, the Bayesian retrieval algorithm performs significantly better than NN zenith also for estimating temperature profile.

## 5. CONCLUSION AND DISCUSSION

This paper describes in detail the Bayesian optimal estimation and the improvements applied to the technique to estimate humidity and temperature profiles with increased accuracy. The technique is applied to the radiometer measurements performed for the month of August 2011 and water vapor density and temperature profiles have been retrieved to show that thermodynamic properties can be retrieved and can be used to detect the temporal changes. The retrieved profiles have been compared with those from the NN zenith method and also with the NOAA reanalysis data which is considered as truth in this




case. The results show that Bayesian optimal estimation using a small background information dataset (50 profiles taken over a period of one month) has better performance than the NN zenith method (which requires a large background dataset taken over 4-5 years as training data). This is because the profiles in the background dataset are temporally and spatially correlated with the measurements performed by the radiometer which in turn provide the variability information associated with the water vapor and temperature profiles.

The Bayesian technique, is an optimal combination of ground-based observations and the related background information, hence it retains the information carried by the background dataset thus providing the variability information in the lower troposphere. This results in the Bayesian optimal estimation achieving the improved retrieval performances throughout the altitude of interest.

Water vapor profiles retrieved using the Bayesian optimal estimation technique compares well with the reanalysis data for 16-August-2011 and 26-August-2011 with differences less than 1.5 $g/m^3$ for the whole profile and for other days the difference is lower than the error observed for NN zenith from ground to 3 km above ground level. The RMS error for Bayesian estimation is less than 0.8 $g/m^3$ from ground to 8 km altitude which is less than those observed for NN zenith. Temperature profiles estimated using Bayesian optimal estimation have been observed to have differences of less than 3 K with the reanalysis data while the NN zenith profiles usually have a difference of 3 K or more for the whole profile. The RMS error shows that Bayesian method has error less than 0.7 K while the NN zenith has error higher than 2 K and increases as the altitude increases.

In these analyses, water vapor density and temperature profiles have been estimated from radiometer measurements. These profiles retrieved using 100 m thick atmospheric layers as well as temporally short but statistically significant background data set taken close to the measurement time have a higher probability of sensing temporal variations in humidity and temperature profiles than those estimated by NN zenith. This is because the background dataset is correlated with the measurements and provides variability information about water vapor or temperature profiles.

**ACKNOWLEDGEMENT**

Authors would like to thank National Oceanic and Atmospheric Administration (NOAA), Earth System Research Laboratory for providing such an useful reanalysis dataset which helped in analyzing the error associated with the estimated profiles. We would also like to thank Dr. Xavier Bosch-Lluis for his important contribution and suggestions.

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



# FIGURES

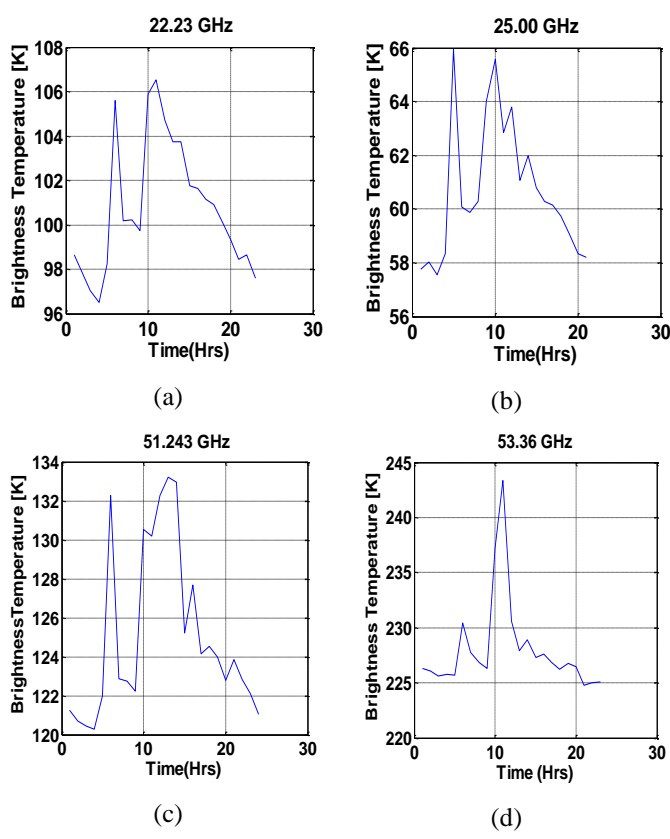

Figure 1. Time series of brightness temperature at 22.23, 25.0, 51.243 and 53.36 GHz.



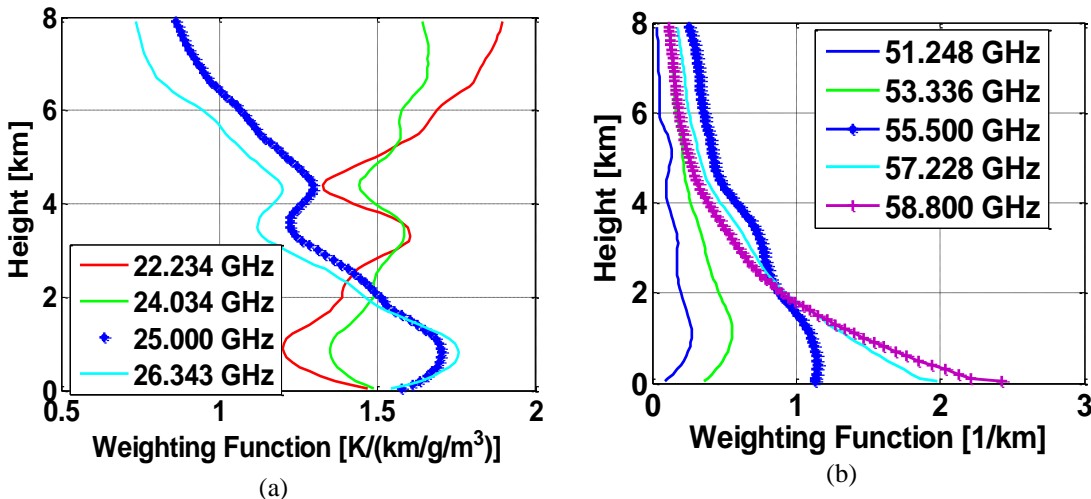

**Figure 2. (a) Weighting functions for measurement frequencies used for water vapor profile retrieval. (b) Weighting functions for measurement frequencies used for temperature profile retrieval.**

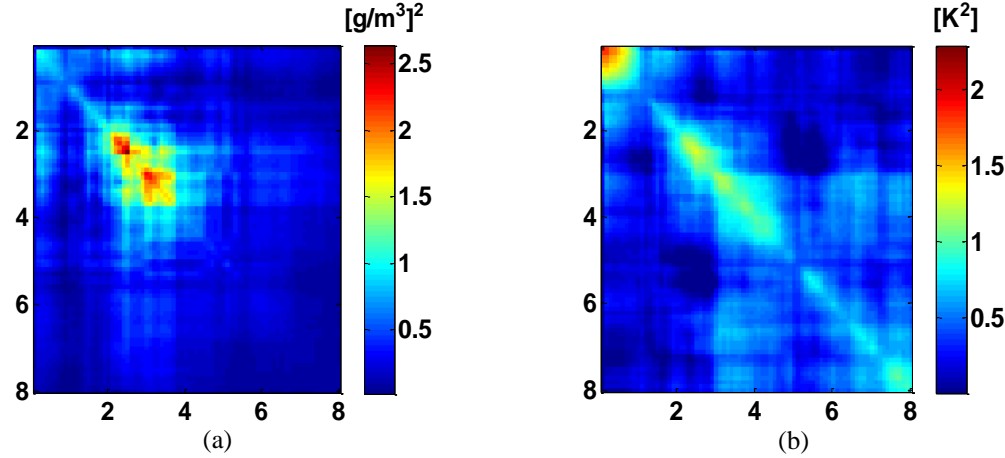

**Figure 3. Background information covariance matrix for 80 layers (100 m thick) (a) water density (b) temperature profiles. The x and y axes are in kilometers for both the figures.**





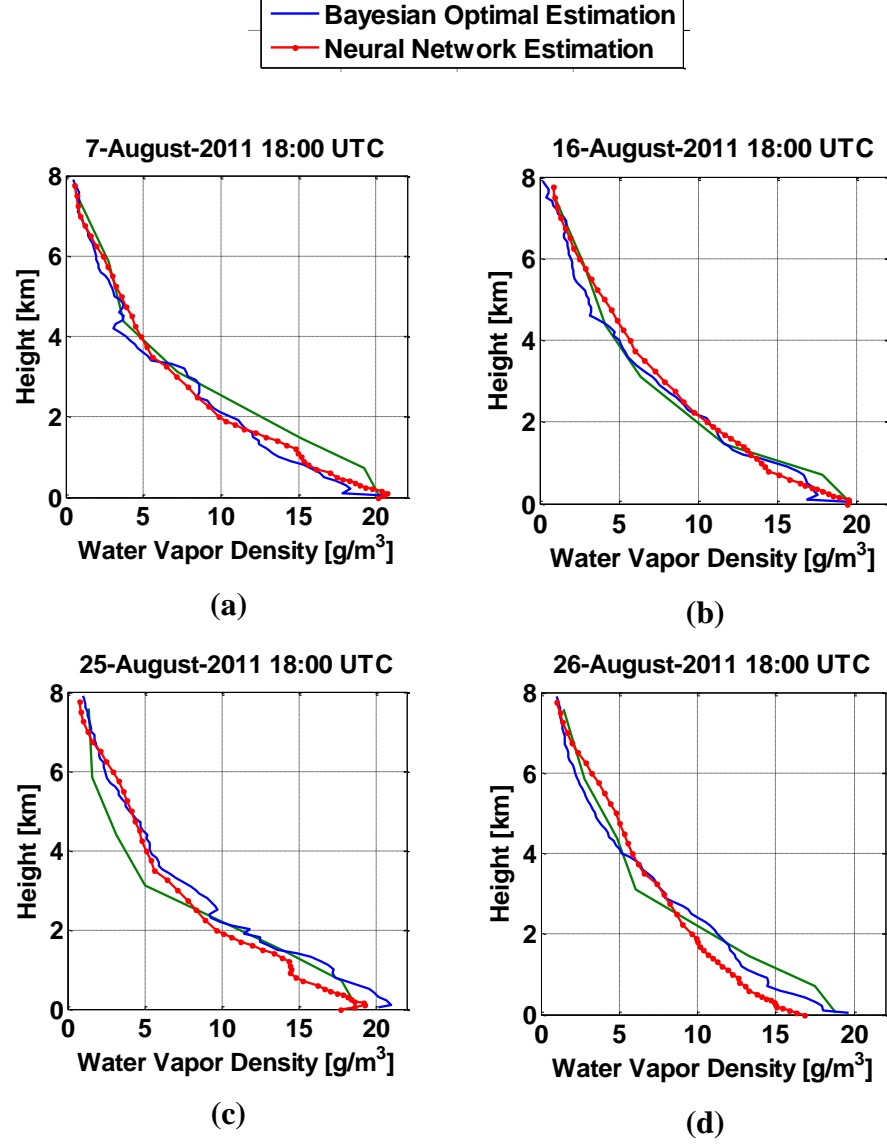

**Figure 4. Time series analysis data of water vapor retrieved profiles.**





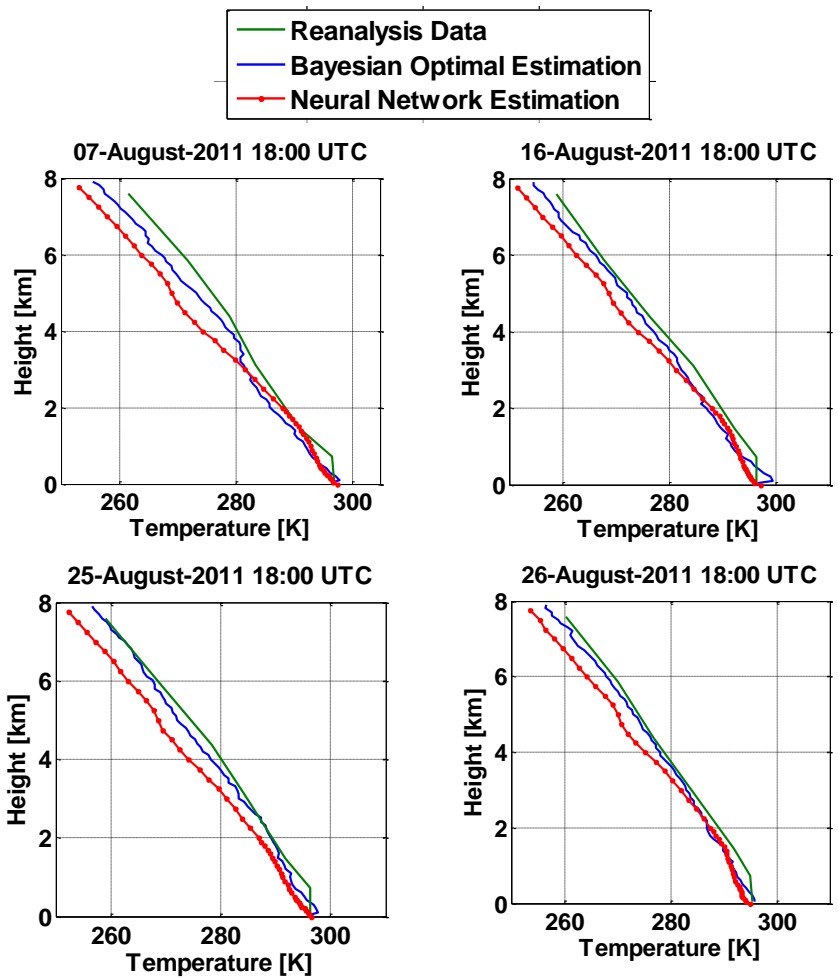

**Figure 5. Time series analysis data of temperature retrieved profiles.**




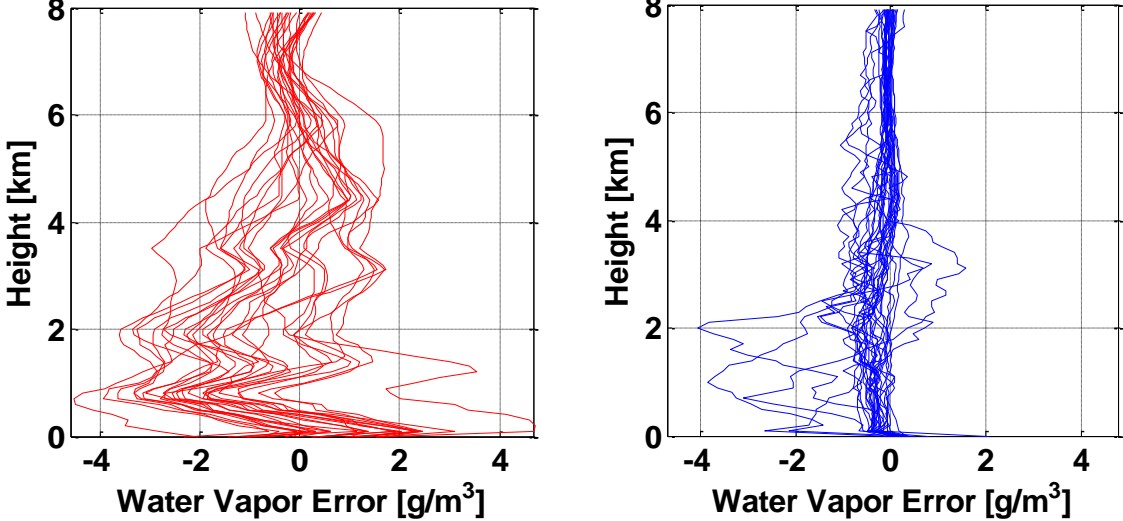

**Figure 6.** Error associated with water vapor density profile retrieved by (a) neural network (b) Bayesian optimal estimation.

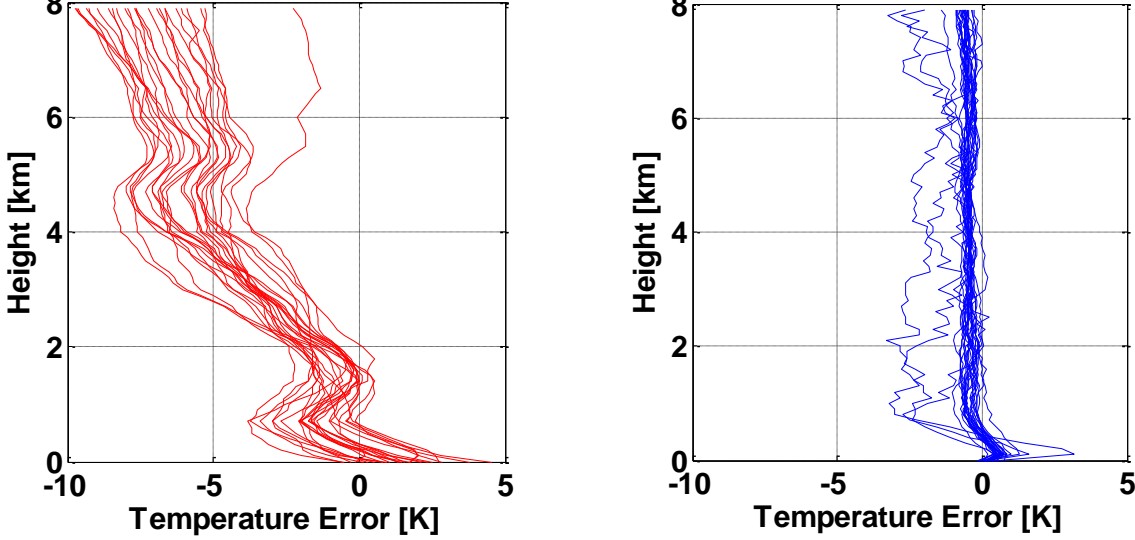

5 **Figure 7.** Error associated with temperature profiles retrieved by (a) neural network (b) Bayesian optimal estimation.





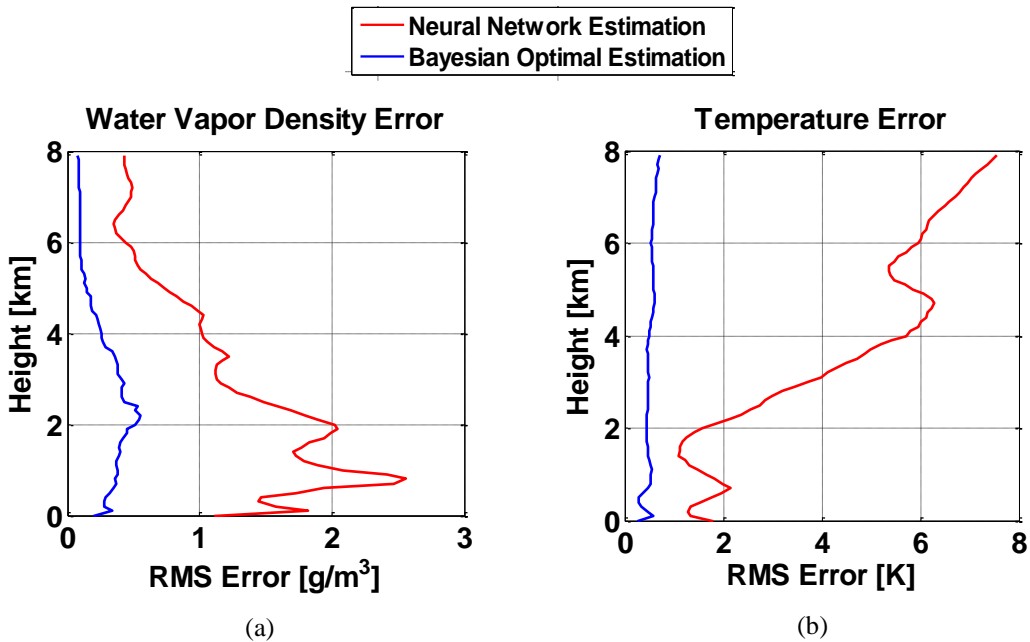

**Figure 8: RMS Error analysis for (a) water vapor profiles and (b) temperature profiles.**