# Peer review of "Time Series Analysis of Ground-Based Microwave Measurements at K- and V-Bands to Detect Temporal Changes in Water Vapor and Temperature Profiles"

_Geoscientific Instrumentation, Methods and Data Systems, 2016_

## Referee Comment (RC1) · Anonymous Referee #2 · 18 Aug 2016

revised the paper before it was posted in the forum. In my opinion the paper can be accepted as it is now.

---

## Short Comment (SC1) · 14 Sep 2016

Expand all of the acronyms e.g. SAPHIR-MADRAS, NN, AMSU, FLORA, MP-3000A. . .

Page 2, line 2: add to the reference list the following: Spuler, S. M., Repasky, K. S., Morley, B., Moen, D., Hayman, M., and Nehrir, A. R.: Field-deployable diode-laser-based differential absorption lidar (DIAL) for profiling water vapor, Atmos. Meas. Tech., 8, 1073-1087, doi:10.5194/amt-8-1073-2015, 2015.

Page 2, line 10: Brogniez, et al.2013 does not show any retrieved humidity or temperature profile. Add an appropriate reference.

[Figure]

Page 2, line 13: What is meant by 'window frequency?'

Page 3, line 18: Define 'oxygen complex.'

Figure 2: Explain why 22 GHz weighting function is lower than the 25 GHz weighting function but TB of 22 > 25 GHz as shown in Figure 1?

Explain why 53 GHz weighting function is lower than the 51 GHz weighting function but TB of 53 > 51 GHz as shown in Figure 1?

Page 4, line 9: Define apriori

Page 4, line 10: Describe 'background information statistics' and how it is used to constrain then inversion.

Figure 3: Add X and Y-axis labels. Why is Y-axis inverted?

Page 4, line 24: Spurious character '5'; something is missing.

Page 4, last line: What is the difference between apriori and background information covariance matrix?

Page 5 line 15: Why values of diagonal elements are fixed as 0.25 K? How is this value determined? How sensitive is this value to RMS errors shown in Figures 7 and 8?

Page 5: How equations 1 and 2 are related?

Page 7, line 9: Do the authors mean inversion instead of the gradient?

Page 7, line 29: Change Section 4C to Section 4.3

Figure 6b: Explain why a few of the water vapor profiles have errors > -2 g/ mˆ2 ?

Page 9, line 21-22: Describe what aspects of background information are correlated with measurements and why they are correlated?
* * *
16, 2016.

---

## Referee Comment (RC2) · Anonymous Referee #3 · 26 Sep 2016

1) To improve the dissemination of the manuscript, please add a paragraph describing the physical phenomenon at the basis of the sensing 2) For the Bayesian Optimal Estimation section: 2.1 If the equation (1) is the updating iterative minimization step of the cost function in eq. (2), it necessary to define first the cost function and after to provide the minimization step. In other words, please eq.(1) becomes eq.(2) and eq.(2) becomes eq. (1). 2.2 How do you choose the value o f m in eq. (3) to define the convergence criterion ? 2.3 Please, comment more in detail about the effect of parameter ÏŠ on the local minima problem. 3) The section about the Neural Network (NN) estimation should be enlarged. What is the number of the unknowns searched

for in the NN estimation ? How long was the time to train the network ? 4) By looking to figure 8, please comment the fact (maybe, by providing a "physical justification") that the RMS error for NN estimation, has a decreasing behaviour with kilometres for the water content profile whereas increases with the distance for the temperature profile.

---

## Author Comment (AC1) · 24 Oct 2016

The authors would like to thank the reviewer for his comments due to which the quality of the paper has improved.
* * *

---

## Author Comment (AC2) · 24 Oct 2016

The response to the reviewer comment is attached

Please also note the supplement to this comment:
http://www.geosci-instrum-method-data-syst-discuss.net/gi-2016-16/gi-2016-16-AC2-supplement.pdf
* * *
[Figure]

*Referee #3*

*Response to Reviewer Comments*

The authors would like to thank the reviewer for the comments which have significantly improved the quality of the paper.

*1) To improve the dissemination of the manuscript, please add a paragraph describing the physical phenomenon at the basis of the sensing.*

**Response:** The authors have added the following paragraph to the paper in Section 3.1.

Remote sensing of water vapor and temperature is based on the measurement of microwave radiation emitted by water vapor and oxygen molecules. The emission and absorption of microwave radiation due to water vapor and oxygen in each tropospheric layer change the microwave radiation that reaches the ground. This variation in radiation is due to the concentration of water vapor in the atmosphere and the temperature at various altitudes. Therefore, these microwave radiation reaching the ground are source of information about the humidity distribution and temperature variation in the atmosphere at different heights.

Measurement of this radiation at the weak water humdity absorption line (centred at 22.235 GHz) is used for the sensing of water vapor profile variation. This is based on humidity absorption line broadening. This broadening is due to motion of the water molecules and their collisions with other water molecules and is known as pressure broadening. Thus change in pressure has a significant impact on the width of the absorption lines as well as the absorption values. So, a decrease in the atmospheric pressure reduces the line width and increases the water vapor absorption line strength which is most prominent at 22.235 GHz (the center of the absorption line). Therefore, closer the proximity of the measurement frequency to the weak water vapor resonance frequency higher the sensitivity to water vapor at high altitudes. As the pressure increases the absorption line widens resulting in reduced sensitivity to water vapor at high altitudes. However, frequencies farther away from the center frequency are more sensitive to water vapor changes close to ground level. This is again proven by the weighting functions at various frequencies. Weighting functions closest to the water vapor resonance frequencies are almost twice more sensitive to water vapor at 8 km than near ground level. While frequencies further way from the resonance peak are most sensitive to changes close to ground level. Therefore, a combination of various frequency measurements is able to detect the profile information about water vapor.

Similarly, microwave radiation from oxygen at the 60 GHz absorption complex can be used for retrieving temperature profile information because atmospheric absorption in the 50-75 GHz range is primarily due to oxygen molecules. The absorption due to oxygen molecule is due to magnetic moment 33 spin-rotational lines between 51.5-67.9 GHz. These spin-rotational lines blend together at lower altitude due to the pressure broadening of the lines. This blended absorption lines has a shape similar to an absorption band centered at 60 GHz. However, the absorption line intensity is not the simple addition of isolated line intensities but the "overlap interference" which gives rise to a very complex absorption band called the

**Fig. 1.** Response to Reviewer comment

**Supplement:**

**Time Series Analysis of Ground-Based Microwave Measurements at K- and V-Bands to Detect Temporal Changes in Water Vapor and Temperature Profiles**

Sibananda Panda[1], Swaroop Sahoo[2], G. Pandithurai[3]

[1]School of Electronics Engineering, KIIT University, Odisha, India
[2]Department of Electrical Engineering, Indian Institute of Technology Palakkad, Palakkad, Kerala, India
[3]Indian Institute of Tropical Meteorology, Pune, India

*Correspondence to*: Swaroop Sahoo (swaroop.sahoo769@gmail.com)

**Abstract.** Ground-based microwave measurements performed at water vapor and oxygen absorption line frequencies are widely used for remote sensing of tropospheric water vapor density and temperature profiles, respectively. This paper focuses on using time series of independent frequency measurements at K- and V-bands along with statistically significant but short background data sets to retrieve and sense temporal variations and gradients in water vapor and temperature profiles. To study this capability, Indian Institute of Tropical Meteorology (IITM) had deployed a microwave radiometer at Mahabubnagar, Hyderabad during August 2011 as part of the Integrated Ground Campaign during the Cloud Aerosol Interaction and Precipitation Enhancement Experiment (CAIPEEX-IGOC). In this study, time series of water vapor and temperature profiles were retrieved using Bayesian optimal estimation method which uses Levenberg-Marquardt optimization technique. The temperature profiles for the first time have been estimated using optimized background information covariance matrix so as to improve the accuracy of the retrieved profiles as well as be able to detect gradients. Estimated water vapor and temperature profiles are compared with those taken from the reanalysis data updated by the Earth System Research Laboratory National Oceanic and Atmospheric Administration (NOAA), to determine the range of possible errors. Similarly, RMS errors are evaluated for the water vapor and temperature profiles for a month to estimate the accuracy of the retrievals. It is found that water vapor and temperature profiles can be estimated with an acceptable accuracy by using a background information data set compiled over a period of one month.

[revised manuscript text omitted]

**2.  INSTRUMENTS DEPLOYMENT**

Indian Institute of Tropical Meteorology (IITM) had deployed a microwave radiometer in Mahabubnagar (16° 44  N, 77° 59 E), Hyderabad for the whole month of August, 2011 as part of the Integrated Ground Campaign during the Cloud Aerosol Interaction and Precipitation Enhancement Experiment (CAIPEEX-IGOC) (Leena et. al., 2015). This is a frequency agile radiometer and operated at 8 frequencies in the range 22-30 GHz and 14 frequencies from 51.0 GHz to 58.0 GHz in V-band, at elevation angles of $15^o$, $90^o$ and $165^o$. The resolution of the instrument varies from 0.1 to 1 K depending on integration time i.e., 0.01 to 2.5 seconds (Radiometrics Corporation, 2008) . The accuracy of the brightness temperature measurements is approximately 0.2 K and the bandwidth of the channels is 300 MHz. This instrument also has a single channel infrared radiometer in addition to surface pressure, humidity and temperature sensors. The multichannel microwave radiometer is calibrated by injecting noise from a noise diode to remove the system gain fluctuations. Two sided tipping curve calibration method has been used to determine the brightness temperatures from the measured voltages for water vapor channels and the cold (liquid nitrogen) and hot load calibration (internal black body at ambient temperature) is used to calibrate the temperature channels measurements.

Radiometer measurements during the field campaign were performed throughout the day and night under varying atmospheric conditions which included clear and cloudy skies. The time series of calibrated brightness temperatures for 22.23, 25.0, 51.243 and 53.36 GHz are shown in Figure 1. It can be observed that brightness temperatures at 22.23 GHz are comparatively higher than those at 25 GHz. This is because 22.23 GHz is the water vapor resonance frequency and is more sensitive to water vapor in the atmosphere than 25 GHz which is significantly far away from the water vapor resonance frequency. Similarly, measurements at 53.36 GHz are higher than those at 51.243 GHz because of the proximity of 53.36 GHz to the oxygen complex. Thus, measurement frequencies are sensitive to water vapor and temperature to a varying extent. This can also be confirmed by analyzing the weighting functions corresponding to water vapor and temperature frequencies shown in Figure 2. Figure 2(a) shows that weighting function values for 22.234 GHz are higher than those at 25.00 GHz at altitudes above 2 km while weighting function values at 25 GHz have higher values than those at 22.234 GHz below 2 km. This is because the measurements at 22.234 GHz are comparatively more sensitive to changes in water vapor at altitudes above 2.5 km while those at 25.00 GHz are more sensitive to changes in water vapor below that altitude. However, the weighting function values at 22.234 GHz for altitude range 2.5-8 km are significantly higher than those at 25.00 GHz so that brightness temperatures at 22.234 GHz are still higher than those at 25.00 GHz.

The temperature measurement frequencies shown in Figure 2 (b) are most sensitive to temperature variations from 0 to 4 km altitude. 53.36 GHz weighting function (represented in green in Figure 2 b) is higher than 51.243 GHz weighting function (represented in blue in Figure 2 b) at all altitudes.

To complement the radiometer measurements, Vaisala RS92-SGP radiosondes were launched everyday at 12 UTC. These radiosondes were launched from the radiometer deployment location to provide vertical profiles (temporal resolution of two seconds) of relative humidity, temperature, dew point temperature, pressure, and wind. These radiosondes data have been used as the source of apriori information as well as the source of background dataset during this study and analysis.

**3. THEORETICAL BACKGROUND**

**3.1 Remote Sensing of Water Vapor and Temperature Profile**

Remote sensing of water vapor and temperature is based on the measurement of microwave radiation emitted by water vapor and oxygen molecules. The emission and absorption of microwave radiation due to water vapor and oxygen in each tropospheric layer change the microwave radiation that reaches the ground. This variation in radiation is due to the concentration of water vapor in the atmosphere and the temperature at various altitudes. Therefore, these microwave radiation reaching the ground are source of information about the humidity distribution and temperature variation in the atmosphere at different heights.

Measurement of this radiation at the weak water humdity absorption line (centred at 22.235 GHz) is used for the sensing of water vapor profile variation. This is based on humidity absorption line broadening. This broadening is due to motion of the water molecules and their collisions with other water molecules and is known as pressure broadening. Thus change in pressure has a significant impact on the width of the absorption lines as well as the absorption values. So, a decrease in the atmospheric pressure reduces the line width and increases the water vapor absorption line strength which is most prominent at 22.235 GHz (the center of the absorption line). Therefore, closer the proximity of the measurement frequency to the weak water vapor resonance frequency higher the sensitivity to water vapor at high altitudes. As the pressure increases the absorption line widens resulting in reduced sensitivity to water vapor at high altitudes. However, frequencies farther away from the center frequency are more sensitive to water vapor changes close to ground level. This is again proven by the weighting functions at various frequencies. Weighting functions closest to the water vapor resonance frequencies are almost twice more sensitive to water vapor at 8 km than near ground level. While frequencies further way from the resonance peak are most sensitive to changes close to ground level. Therefore, a combination of various frequency measurements is able to detect the profile information about water vapor.

Similarly, microwave radiation from oxygen at the 60 GHz absorption complex can be used for retrieving temperature profile information because atmospheric absorption in the 50-75 GHz range is primarily due to oxygen molecules. The absorption due to oxygen molecule is due to magnetic moment 33 spin-rotational lines between 51.5-67.9 GHz. These spin-rotational lines blend together at lower altitude due to the pressure broadening of the lines. This blended absorption lines has a shape similar to an absorption band centered at 60 GHz. However, the absorption line intensity is not the simple addition of

isolated line intensities but the "overlap interference" which gives rise to a very complex absorption band called the oxygen complex. As a result the opacity at the 60 GHz is significantly higher than that at 50 GHz, so the radiometer just observes the radiation emitted close to the ground surface. To sample the whole troposphere measurements need to be performed at a number of frequencies away from the center frequency.

5   Since, oxygen is the most well mixed gas in the atmosphere and its proportion in the atmosphere is almost constant and altitude independent from ground level to 80 km, the microwave radiation at the oxygen absorption lines contains atmospheric temperature profile information.

**3.2 Retrieval Techniques**

10  **3.2.1    Bayesian Optimal Estimation**

The Bayesian optimal estimation is an inversion method using multiple K- and V-band microwave frequency measurements to retrieve profiles of humidity and temperature. This is because various measurement frequencies have varying sensitivity to water vapor and temperature at various altitudes. These multiple measurements combined with the a-priori information and a background information covariance matrix can be used for retrieving water vapor and temperature profiles while sensing the

15  associated temporal variations and gradients.

The retrieval of water vapor and temperature profiles from brightness temperature measurements is a non-linear problem; a variation of the iterative Gauss-Newton method is required for the estimation process. This retrieval technique uses a-priori humidity and temperature information as a constraint to determine an unique solution to the inverse problem. A-priori in this paper represents the measurement of water vapor and temperature profiles prior to the radiometer brightness temperature

20  measurements. This is also known as the initialization profile in this paper. The a-priori information is taken from radiosonde launched a few hours before the radiometer performs the measurement.

In addition to the a-priori information, water vapor density and temperature background information statistics is also introduced as the inversion constraint. These statistics provide variability information associated with the atmospheric humidity and temperature profiles as well as the inter layer correlation during a particular time period. The number of

25  elements in the background data set and the relationships among them determines the values of the background information covariance matrix elements. Since, in this study the dataset used for calculating the background statistics has been taken close to measurement time it will be more representative of weather conditions during that time period and location. Background information statistics here means the background information covariance information represented by the matrix $\overline{\overline{S}}_a$.

30  The Bayesian optimal estimation uses the Levenberg-Marquardt (LM) optimization method (Rodgers, 2000). The LM technique is shown in Eqs. (1).

$$\overline{x}_{i+1} = \overline{x}_i + \left((1+\gamma)\overline{\overline{S}}_a^{-1} + \overline{\overline{K}}_i^T \overline{\overline{S}}_\epsilon^{-1} \overline{\overline{K}}_i\right)^{-1} \left(\overline{\overline{K}}_i^T \overline{\overline{S}}_\epsilon^{-1}[\overline{T}_E^r - \overline{T}_E(\overline{x}_i)] - \overline{\overline{S}}_a^{-1}[\overline{x}_i - \overline{x}_a]\right) \tag{1}$$

where i is the iteration index, $\overline{K}_i$ is the kernel or weighting function matrix and determines the sensitivity of the measurements at various frequencies to changes in the parameter of interest at various altitudes, $\overline{x}_i$ is the water vapor density or temperature profile which is updated at each iteration and is same as initialization profile for i =1, $\overline{T}_B^r$ is the measured brightness temperature vector at water vapor density or temperature measurement frequencies, $\overline{x}_a$ is the background profile and is same as the initialization profile in this case because a small dataset is used as background dataset. $\overline{T}_B(\overline{x}_i)$ is the radiative transfer model simulated brightness temperature using the absorption coefficients calculated from a Rosenkranz model (Rosenkranz, 1993) (Rosenkranz, 1998).

$\overline{S}_\epsilon$ is the observation error covariance matrix and contains the uncertainty information associated with the measurement. The observation error covariance matrix takes into consideration the radiometric measurement noise, representativeness error and radiative transfer model errors. The diagonal elements of the observation error covariance matrix are approximately in the range of 0.23 to 0.29 $K^2$ and some of the off-diagonal elements are close to zero. The observation error covariance, R, determines the uncertainty associated with the observations. This uncertainty has contributions from radiometric noise (E), forward model (F) and representativeness (M) errors. Radiometric noise is determined based on radiometric resolution which is the minimum difference in scene brightness temperature that can be sensed by the receiver. This value for MP3000-A varies from 0.1 to 1 K depending on integration time (Radiometrics Corporation, 2008). The typical value is 0.25 K for each measurement frequency while considering an integration time of 250 msec. In addition to the radiometric noise, forward model errors introduced due to inadequate absorption models. These are determined using the difference between brightness temperatures simulated by two absorption models i.e., the Rosekranz model as well as MPM93 (Liebe et al., 1993) which are then used to determine the forward model error covariance matrix. Another source of uncertainty is the representativeness error which takes into consideration the radiometer's sensitivity to fluctuations in the atmosphere on a time scale shorter than that can be represented by any numerical weather prediction model or radiosondes profiles. The representative covariance is calculated in Eq. 2.

$$M = E(\overline{T}_B^r(t + \Delta t) - \overline{T}_B^r(t))(\overline{T}_B^r(t + \Delta t) - \overline{T}_B^r(t))^T \qquad (2)$$

where t is time and $\Delta t$ is the time scale of difference. The observation error covariance matrix is shown in Figure 3 and given by $R = E + F + M$.

$\overline{S}_a$ is the background covariance matrix which is computed using information from 50 radiosonde profiles launched over a period of one month. $\gamma$ is the LM factor and the value of $\gamma$ is updated at each iteration based on value of $J(x)$ from Eq. (3). Various initial values of $\gamma$ in the range of $\gamma = 1$ and $\gamma = \infty$ have been considered for starting of the iteration. For $\gamma = 1$, the iteration might move towards a local minima while in case of $\gamma=$ the iteration immediately moves towards the global minima which gives a solution which does not converge. Therefore, the initial value of $\gamma$ is assumed to be one. It was observed that the algorithm did not converge with a valid output for this initial value of gamma so the initial value of gamma is increased at regular intervals to check the convergence. It was found empirically, that gamma with an initial value of 5000 converges the algorithm for all cases. As part of the iteration if the value of $J(x)$ increases, then the iteration is discarded and

the value of γ is increased 10 fold and the iteration is repeated. This is done so as to discard any invalid output which could be close to one of the local minima. If value of J(x) decreases, then the iteration is valid and the value of γ is reduced by a factor of 2 for the next iteration even if the convergence criteria is not satisfied (Hewison, 2007). This process is followed until the convergence criterion is validated by the output profile. This process is illustrated in Figure 4. It can be observed that as the cost function decreases, the gamma value decreases and vice versa. At local minima of the cost function the gamma value also reduces.

LM technique output is dependent on the cost function represented by J(x) in Eq. (3)

$$J(x) = [\bar{x} - \bar{x}^b]^t \bar{\bar{S}}_{11}^{-1} [\bar{x} - \bar{x}^b] + [\bar{T}_B(\bar{x}_i) - \bar{T}_B^r]^t \bar{\bar{S}}_E^{-1} [\bar{T}_B(\bar{x}_i) - \bar{T}_B^r] \tag{3}$$

where $\bar{x}^b$ and $\bar{x}$ are the initialization profiles (either water vapor or temperature) and output profiles (either water vapor or temperature) for each iteration, respectively. The final water vapor or temperature output profiles are determined by the convergence criterion given by Eq. (4)

$$[\bar{T}_B(\bar{x}_{i+1}) - \bar{T}_B(\bar{x}_i)]^t \bar{\bar{S}}_\delta^{-1} [\bar{T}_B(\bar{x}_{i+1}) - \bar{T}_B(\bar{x}_i)] \ll m \tag{4}$$

where $m$ is 5 and 7 (dimension of water vapor and temperature measurement vector) for water vapor and temperature profile retrieval and $\bar{\bar{S}}_\delta$ is the covariance between $\bar{T}_E^r$ and $\bar{T}_B(\bar{x}_i)$. Eq. (4) determines the termination of the iterative process. The iteration stops when Eq. (4) reaches a value q which is very small in comparison to m. Therefore, the value of q is chosen to be 0.05 and 0.07 for water vapor and temperature profile retrieval, respectively, which is 1/100 times the number of measurements used).

**3.2.2 Impact of Background Dataset on Retrieval**

As already studied and determined by Scheve (Scheve & Swift, 1999), Hewison(Hewison, 2007), Solheim (Solheim, et al., 1998) and Sahoo (Sahoo et. al., 2015a) the number of measurement frequencies which provide altitude related information about water vapor and temperature are limited by the information content or degrees of freedom of the measurements. Thus, use of these measurements in various inversion methods to retrieve the thermodynamic properties at more number of altitudes than the information content limit is an ill-posed as well as a non-linear problem. To overcome these short comings Bayesian optimal estimation method uses background information statistics.

Background information statistics here means the background information covariance information represented by the matrix $\bar{\bar{S}}_a$ and determines the range of variability information associated with water vapor or temperature profiles during a time period. Background data set is very important for the performance of the retrieval algorithm in terms of accuracy and ability to sense temporal changes. This is because background data set taken closer in time to the radiometer measurement will describe the atmospheric conditions i.e., the temperature and humidity profiles as well as the wind vector etc. during the measurements. So, the inversion of the measurements results in retrieval of the most persistent water vapor and temperature profile while being constrained by the background data set and a-priori. As a result, the retrieved profiles will detect similar features as the actual profile and hence will represent the gradients and dynamic changes in the actual profile. The

atmospheric conditions during a particular season or month are correlated because the atmospheric conditions are similar throughout the time period accept a few outliers which cannot be correlated to the time of interest. Therefore, measurements along with the background data set and the a-priori will retrieve the most probable water vapor and temperature profile while an outlier might or might not be detected depending on whether that event is properly described by the background covariance matrix.

The background information covariance matrix used in this paper is shown in Figure 5 (a) and (b) and are calculated using water vapor density and temperature profiles measured over a period of one month, respectively. It can be observed that most of the water vapor variability information is between 20-40 layers which correspond to the altitude range of 2 to 4 km. On the other hand, Figure 5 (b) shows that the temperature variability information is primarily below an altitude of one km and also in the range of 2-4.2 km. In contrast to these results, when background information covariance matrix is computed from a large dataset, important weather events or temporally varying conditions are overshadowed because the covariance matrix takes into the consideration the overall variability information while reducing the weight of certain weather conditions which correspond to a particular season (Sahoo et.al., 2015b).

The goal of this study is to retrieve water vapor and temperature profiles with improved accuracy while using a background dataset measured over a period of one month so as to detect the temporal changes and gradients in the lowest 8 km of the profiles.

**3.3 Neural Network Estimation**

Estimation of water vapor and temperature profiles from microwave radiometer brightness temperatures is done using a proprietary neural network method (NN) developed by Radiometrics Corporation (Solheim, et al., 1998). NN zenith estimation of temperature, water vapor density, relative humidity, and liquid water content profiles are performed simultaneously from the radiometer measurements plus the infrared (IR) channel. The retrieved profiles are estimated at 58 height levels, with 50 meter steps from the surface up to 500 m, then 100 m steps to 2 km, and 250 m steps from 2 to 10 km. However, it has to be noted that above approximately 7 km, the atmospheric water vapor density and temperature approach the climatological mean values.

As part of the retrieval process the NN is trained using a back-propagation algorithm and radiosonde data which has been collected over a period of time i.e., usually 4 to 5 years using. The radiosonde data that is used for training the network is taken from one or more sites with the same climatological conditions as the observation site. The radiosonde profiles are used for simulating the brightness temperature using absorption models and radiative transfer equations. The NN estimation uses a standard feed-forward network (Radiometrics Corporation, 2008) to retrieve the temperature, humidity and liquid water profile that is most consistent with the atmospheric conditions and radiometric measurements.

[revised manuscript text omitted]
). In addition to that, it can be observed that some of the retrieved profiles in Figure 8 (b) showed higher than usual errors i.e., 2 $g/m^2$ and above. This is because the water vapor profile retrieval accuracy is significantly affected by the a-priori profile as shown by Sahoo et al., 2015. If the atmospheric conditions during the a-priori profile measurement (radiosonde launch) are very different from the conditions during the radiometer measurements then the actual profile will be different from the a-priori. This will result in errors which are higher than when the a-priori and estimated profiles are similar or the weather conditions for the two times are not very different. This difference in weather conditions is due to a weather phenomenon or a rain event.

The range of errors associated with the temperature profiles are shown in Figure 9 (a) and (b) for both the NN and Bayesian optimal techniques, respectively. The error associated with the neural network profile is in the range of -3 to 5 K in the lowest 1 km of the troposphere and then the range changes to -4 to -8 K at 4 km above ground level. It is clear that the NN zenith retrieval underestimates the value of the temperature profile. The error associated with the Bayesian optimal estimated profile is shown in Figure 9 (b) and is in the range of -1 to 0 K except in the case of a few profiles. As in the case of water vapor profile, the errors associated with temperature profiles by Bayesian optimal estimation are significantly less than NN estimated profiles. The Bayesian optimally estimated retrievals using radiometer observations compare well with the reanalysis data in the lowest 2 km of troposphere because of the retrieval being constrained by apriori and surface measurements provided by the radiometer.

In addition to the above analysis another analysis was performed to determine the sensitivity of retrieved profile to changes in observation error covariance matrix in the Bayesian optimal estimation. Water vapor and temperature profiles have been already retrieved for various days as shown in Figure 6 and Figure 7. The profile retrieved for 7-August-2011 has been reanalyze where the profile is retrieved using the observation covariance matrix shown in Figure 3. Again the profile is retrieved after increasing the diagonal element of the observation error covariance matrix by 0.25 $K^2$. The retrieved profiles for water vapor and temperature profile for both the observation error covariance matrices are shown in Figure 10. It can be observed that the retrieved water vapor profile for the new covariance matrix has higher error than the profile for the covariance matrix shown in Figure 3. The increase in error for the retrieved water vapor profile is in the range of 0.3 to 1.9 $g/m^3$ (0 to 8 km altitude) as the diagonal elements are increased by 0.25 $K^2$. Similarly, the increase of 0.25 $K^2$ in the diagonal elements of the temperature observation covariance matrix shown in Figure 3 increases the temperature error by 0.2 to 0.5 K (0 to 8 km altitude). Thus, the observation error covariance matrix has a significant impact on the retrieved profile quality and accuracy.

Another analysis was performed to determine the deviation of the retrieved profiles from the reanalysis profile, root mean squared (RMS) errors are calculated for both the methods. RMS errors are calculated by comparing radiometer retrieved humidity and temperature profiles (retrieved using both Bayesian optimal estimation and NN method) with the reanalysis data (which is used as truth in this case) and are shown in Figure 11. Figure 11 (a) shows that RMS error associated with Bayesian optimal estimated water vapor profile varies from 0.2 to 0.4 $g/m^3$ for the lowest 4 km of the troposphere and is less than 0.2 $g/m^3$ above 5 km altitude. On the other hand, RMS error for NN retrieved profile is in the range of 1-2.5 $g/m^3$ in the lowest 2 km and is less than 1 $g/m^3$ above 4 km. Thus, the RMS error for water vapor profile retrieved using Bayesian optimal estimation is better than NN.

It can be observed from Figure 11 (b) that the RMS error for Bayesian optimal estimated temperature profiles is less than 0.6 K at any altitude from 0 to 8 km above ground level. However, the NN zenith retrieval error range is 1-2 K for lowest 2 km and then increases consistently above 2 km. The maximum error is approximately 7.5 K at 8 km above ground level. Thus, the Bayesian retrieval algorithm performs significantly better than NN zenith also for estimating temperature profile.

It has to be observed that that the RMS error for NN estimated water vapour density profile has a decreasing behaviour with altitude whereas the temperature profile RMS error increases with the height for the temperature profile. This is because NN algorithm used to retrieve the water vapor and temperature profiles has been trained using a data set which has been taken from areas which had similar weather conditions as the radiometer observation site. However, two sites at the same altitude and longitude may have significantly different weather depending on the general conformation of the mountains in the area, the marine currents as well as the advection processes. This could lead to biases in the training of the radiometer algorithm which in turn would increase the error of the retrieved profile. This is what is causing the retrieval errors for both the water vapor and temperature profiles at the lower altitudes. However, at high altitudes the range of water vapor density values which are possible are limited and close to zero as they cannot be less than zero, (obviously the climatological mean) due to which the error levels reduce at high altitudes as shown in Figure 11(a). This is not the case for temperature profiles which can have really low values at high altitudes based on training data. Therefore, the temperature profiles have a high level of bias, hence the increase in error as the altitude increases.

**5. CONCLUSION AND DISCUSSION**

This paper comprehensively describes the Bayesian optimal estimation and the improvements applied to the technique to estimate humidity and temperature profiles with increased accuracy. The Bayesian technique, is an optimal combination of ground-based observations and the related background information, hence it retains the information carried by the background dataset which provides the variability information in the lower troposphere. The background dataset is one of the important parameters in improving accuracy whose effect has been studied in great detail. The technique is applied to the radiometer measurements performed for the month of August 2011 to retrieve water vapor density and temperature profiles. The retrieved profiles show that gradients can be detected along with temporal changes. The retrieved profiles have been compared with those from the NN method and also with the NOAA reanalysis data which is considered as truth in this case. The results show that Bayesian optimal estimation using a small background information dataset (50 profiles taken over a period of one month) has better performance than the NN method (which requires a large background dataset taken over 4-5 years as training data) when a large background data set is not available to train the algorithm. This is because the profiles in the background dataset are temporally and spatially correlated with the measurements performed by the radiometer. Thus, the most persistent profile is retrieved and the Bayesian optimal estimation achieves the improved retrieval performances throughout the altitude of interest.

Water vapor profiles retrieved using the Bayesian optimal estimation technique (Figure 6) compares well with the reanalysis data for 16-August-2011 and 26-August-2011 with differences less than 1.5 $g/m^3$ for the whole profile and for other days the difference is lower than the error observed for NN from ground to 3 km altitude. For most of the days the absolute errors are less than 2 $g/m^3$. In addition to that retrieved profiles are able to detect the gradients in the water vapor profile which are smoothed by the reanalysis data. On the other hand the RMS error for Bayesian estimation is less than 0.8 $g/m^3$ from ground to 8 km altitude which is less than those observed for NN. So the water vapor profile can be retrieved with an accuracy of

better than 1.5 g/m$^3$. Temperature profiles estimated using Bayesian optimal estimation have been observed in Figure 7 to have differences of less than 3 K when compared with the reanalysis data while the NN profiles usually have a difference of 3 K or more for the whole profile. However, on most of the days temperature profiles can be retrieved at an accuracy of better than 1.5 K while detecting the gradients. This has been again proved in the RMS error analysis in Figure 11. The RMS error shows that Bayesian method has error less than 0.7 K while the NN has error higher than 2 K and increases as the altitude increases.

Along with other analyses, one has been performed to determine the sensitivity of retrieved profile accuracy to change in observation error covariance matrix. It has been observed that water retrieved profile error increases by almost 1-2 g/m$^3$ with an increase of 0.25 K$^2$ of the diagonal elements of the matrix. However, an increase of 0.25 K$^2$ of the diagonal elements of the temperature error covariance matrix results in an increase of error less than 0.8 K.

**ACKNOWLEDGEMENT**

Authors would like to thank Earth System Research Laboratory, National Oceanic and Atmospheric Administration (NOAA), for providing such an useful reanalysis dataset which helped in analyzing the error associated with the estimated profiles. We would also like to thank Dr. Xavier Bosch-Lluis for his important contribution and suggestions.

**FIGURES**

[Figure]

(a)  (b)

(c)  (d)

**Figure 1.Time series of brightness temperature at 22.23, 25.0, 51.243 and 53.36 GHz.**

[Figure]

(a)              (b)

**Figure 2. (a) Weighting functions for measurement frequencies used for water vapor profile retrieval. (b) Weighting functions for measurement frequencies used for temperature profile retrieval.**

[Figure]

(a)              (b)

**Figure 3. The observation error covariance matrix for (a) water vapour frequency measurements (b) temperature measurements.**

[Figure]

**Figure 4. The value of cost function and gamma with respect to number of iterations are shown in the top and bottom figure respectively.**

[Figure]

| (a) | (b) |

5   **Figure 5. Background information covariance matrix for 80 layers (100 m thick) (a) water density (b) temperature profiles. The x and y axes are in kilometers for both the figures.**

[Figure]

[Figure]

**Figure 6. Time series analysis data of water vapor retrieved profiles.**

[Figure]

**Figure 7. Time series analysis data of temperature retrieved profiles.**

[Figure]

**Figure 8.** Error associated with water vapor density profileretrieved by(a) neural network (b) Bayesian optimal estimation.

[Figure]

**Figure 9. Error associated with temperature profilesretrieved by(a) neural network(b) Bayesian optimal estimation.**

[Figure]

(a)                                                    (b)

**Figure 10. Retrieved profile sensitivity to observation error covariance matrix (a) Water vapor profile (b) Temperature profile**

[Figure]

(a)                                                    (b)

**Figure 11: RMS Error analysis for (a) water vapor profiles and (b) temperature profiles.**

---

## Author Comment (AC5) · 24 Oct 2016

The authors would like to thank Dr. J. Vivekanandan for the comments. These comments have been very helpful to the authors in increasing the clarity of the paper to the reader.

Below are the comments and the response to the comments

Comment: Expand all of the acronyms e.g. SAPHIR-MADRAS, NN, AMSU,FLORA,MP-3000A... Response: This has been fixed in the paper.
Comment: Page 2, line 2: add to the reference list the following: Spuler, S. M., Repasky, K. S., Morley, B., Moen, D., Hayman, M., and Nehrir, A. R.: Field-deployable diode-laserbased differential absorption lidar (DIAL) for profiling water vapor, Atmos. Meas. Tech., 8, 1073-1087, doi:10.5194/amt-8-1073-2015, 2015. Page 2, line 10: Brogniez, et al.2013 does not show any retrieved humidity or temperature profile. Add an appropriate reference.

Response: The reference (Spuler et. al., 2015) suggested by the reviewer has been added to Section 1 page 2 line 5 of the paper. In addition to that (Brogniez, et al.2013) has been replaced by Rao, T. N., Sunilkumar, K., and Jayaraman, A.: Validation of humidity profiles obtained from SAPHIR, on-board Megha-Tropiques, Special Section: Megha-Tropiques, Current Sci. 104(12), 1635-1642, June 2013 on page 2 line 14.

Comment: Page 2, line 13: What is meant by 'window frequency?'

Response: The window frequency here means the frequency range between the absorption lines (or the peaks) where the atmosphere is transparent to microwave radiation and allows the microwave radiation to pass through without significant attenuation. For example frequency ranges of 30-45 GHz, 70-110 GHz and 125-150 GHz are usually referred to as the window frequency ranges. The window frequencies are still affected by water vapor content and oxygen absorption but are not as sensitive to as the absorption line peaks.

Comment: Page 3, line 18: Define 'oxygen complex.'

Response: The details of the oxygen complex have been added to Section 3.1 line 29 of page 4. "Similarly, microwave radiation from oxygen at the 60 GHz absorption complex can be used for retrieving temperature profile information because atmospheric absorption in the 50-75 GHz range is primarily due to oxygen molecules. The absorption due to oxygen molecule is due to magnetic moment 33 spin-rotational lines between 51.5-67.9 GHz. These spin-rotational lines blend together at lower altitude due to the pressure broadening of the lines. This blended absorption lines has a shape

similar to an absorption band centered at 60 GHz. However, the absorption line intensity is not the simple addition of isolated line intensities but the "overlap interference" which gives rise to a very complex absorption band called the oxygen complex. As a result the opacity at the 60 GHz is significantly higher than that at 50 GHz, so the radiometer just observes the radiation emitted close to the ground surface. To sample the whole troposphere measurements need to be performed at a number of frequencies away from the center frequency."

Comment: Figure 2: Explain why 22 GHz weighting function is lower than the 25 GHz weighting function but TB of 22 > 25 GHz as shown in Figure 1?

Response: This has been rectified and the water vapor weighting function figure has been replaced with an appropriate one as shown in Figure 2(a). The details of the discussion have been added to Section 2 line 27 of page 3. "Figure 2(a) shows that weighting function values for 22.234 GHz are higher than those at 25.00 GHz at altitudes above 2 km while weighting function values at 25 GHz have higher values than those at 22.234 GHz below 2 km. This is because the measurements at 22.234 GHz are comparatively more sensitive to changes in water vapor at altitudes above 2.5 km while those at 25.00 GHz are more sensitive to changes in water vapor below that altitude. However, the weighting function values at 22.234 GHz for altitude range 2.5-8 km are significantly higher than those at 25.00 GHz so that brightness temperatures at 22.234 GHz are still higher than those at 25.00 GHz."

Comment: Explain why 53GHz weighting function is lower than the 51GHz weighting function but TB of 53 > 51 GHz as shown in Figure 1?

Response: The authors would like to clarify and bring to the notice of reviewers that 53.36 GHz weighting function (represented in green in Figure 2 b) is higher than 51.243 GHz weighting function (represented in blue in Figure 2 b) at all altitudes. Therefore, the brightness temperature corresponding to 53.36 GHz is significantly higher than 51.243 GHz.

Comment: Page 4, line 9: Define apriori.

Response: This has been added to Section 3.2.1 line 18 of page 5. "A-priori in this paper represents the measurement of water vapor and temperature profiles prior to the radiometer brightness temperature measurements. This is also known as the initialization profile in this paper. The a-priori information is taken from radiosonde launched a few hours before the radiometer performs the measurement."

Comment: Page 4, line 10: Describe 'background information statistics' and how it is used to constrain the inversion.

Response: This has been added to Section 3.2.1 line 22 of page 5.

"In addition to the a-priori information, water vapor density and temperature background information statistics is also introduced as the inversion constraint. These statistics provide variability information associated with the atmospheric humidity and temperature profiles as well as the inter layer correlation during a particular time period. The number of elements in the background data set and the relationships among them determines the values of the background information covariance matrix elements. Since, in this study the dataset used for calculating the background statistics has been taken close to measurement time it will be more representative of weather conditions during that time period and location. Background information statistics here means the background information covariance information represented by the matrix S ̣_a."

Comment: Figure 3: Add X and Y-axis labels. Why is Y-axis inverted? Response: Figure 3 has become Figure 5. The axes have been fixed in the manuscript.

Comment: Page 4, line 24: Spurious character '5'; something is missing. Response: This has been fixed in the manuscript.

Comment: Page 4, last line: What is the difference between apriori and background information covariance matrix? Response: Apriori here is the observation of the state vector (water vapor and temperature profile) prior to the radiometric measurement. It is

also known as the initialization profile in this paper. Background information statistics here means the background information covariance information represented by the matrix S ݣ_a and determines the range of variability information associated with water vapor or temperature profiles during a time period represented by the background data set. It is calculated using a background data set, which is typically a collection of water vapor or temperature profiles measured over a specific time period and location. Since this matrix provides the variability information it is used to constrain the range of variation in the water vapor or temperature profile.

Comment: Page5 line15: Why values of diagonal elements are fixed as 0.25K? How is this value determined?

Response: The details of observation error covariance matrix have been added to Section 3.2.1 line 8 of page 6.

"S İ£_Ïţ is the observation error covariance matrix and contains the uncertainty information associated with the measurement. The observation error covariance matrix takes into consideration the radiometric measurement noise, representativeness error and radiative transfer model errors. The diagonal elements of the observation error covariance matrix are approximately in the range of 0.23 to 0.29 K2 and some of the off-diagonal elements are close to zero. The observation error covariance, R, determines the uncertainty associated with the observations. This uncertainty has contributions from radiometric noise (E), forward model (F) and representativeness (M) errors. Radiometric noise is determined based on radiometric resolution which is the minimum difference in scene brightness temperature that can be sensed by the receiver. This value for MP3000-A varies from 0.1 to 1 K depending on integration time (Radiometrics Corporation, 2008). The typical value is 0.25 K for each measurement frequency while considering an integration time of 250 msec. In addition to the radiometric noise, forward model errors introduced due to inadequate absorption models. These are determined using the difference between brightness temperatures simulated by two absorption models i.e., the Rosekranz model as well as MPM93 (Liebe et al., 1993)

[Figure]

which are then used to determine the forward model error covariance matrix. Another source of uncertainty is the representativeness error which takes into consideration the radiometer's sensitivity to fluctuations in the atmosphere on a time scale shorter than that can be represented by any numerical weather prediction model or radiosondes profiles. The representative covariance is calculated in Eq. 2. M=E(T ÌĚ_B^' (t+△t)-T ÌĚ_B^' (t)) (T ÌĚ_B^' (t+△t)-T ÌĚ_B^' (t))^T (2) where t is time and △t is the time scale of difference. The observation error covariance matrix is shown in Figure 3 and given by R=E+F+M."

Comment: How sensitive is this value to RMS errors shown in Figures 7 and 8?

Response: The authors assume that the reviewer wants to know the sensitivity of the retrieval errors to measurement error covariance matrix. Figure 7 has become Figure 9 and Figure 8 has become Figure 11. The explanation has been added to Section 4.3 Page 11 line 10. In addition, another analysis was performed to determine the sensitivity of retrieved profile to changes in observation error covariance matrix in the Bayesian optimal estimation. Water vapor and temperature profiles have been already retrieved for various days as shown in Figure 6 and Figure 7. The profile retrieved for 7-August-2011 has been reanalyze where the profile is retrieved using the observation covariance matrix shown in Figure 3. Again the profile is retrieved after increasing the diagonal element of the observation error covariance matrix by 0.25 K2. The retrieved profiles for water vapor and temperature profile for both the observation error covariance matrices are shown in Figure 10. It can be observed that the retrieved water vapor profile for the new covariance matrix has higher error than the profile for the covariance matrix shown in Figure 3. The increase in error for the retrieved water vapor profile is in the range of 0.3 to 1.9 g/m3 (0 to 8 km altitude) as the diagonal elements are increased by 0.25 K2. Similarly, the increase of 0.25 K2 in the diagonal elements of the temperature observation covariance matrix shown in Figure 3 increases the temperature error by 0.2 to 0.5 K (0 to 8 km altitude). Thus, the observation error covariance matrix has a significant impact on the retrieved profile

quality and accuracy.

Comment: Page 5: How equations 1 and 2 are related?

Response: Eq. (2) has become Eq. (3). Eq. (1) is the first step to start the iterative process to determine the water vapor and temperature profiles which are then provided as input to the cost function Eq. (3) as well as the convergence criterion as input for checking the validity.

Comment: Page 7, line 9: Do the authors mean inversion instead of the gradient?

Response: The authors mean the significant changes in the water vapor profiles with respect to altitude in the lowest 4 km as shown in the figures. The word inversion is not used because it is not clear whether it is inversion or just change in water vapor.

Comment: Page 7, line 29: Change Section 4C to Section 4.3 Response: This has been fixed in the paper.

Comment: Figure 6b: Explain why a few of the water vapor profiles have errors > -2 g/ mËĘ2?

Response: Figure 6 has become Figure 8. This response has been added to Section 4.3 page 10 line 27. "In addition to that, it can be observed that some of the retrieved profiles in Figure 8 (b) showed higher than usual errors i.e., 2 g/m2 and above. This is because the water vapor profile retrieval accuracy is significantly affected by the a-priori profile as shown by Sahoo et al., 2015. If the atmospheric conditions during the a-priori profile measurement (radiosonde launch) are very different from the conditions during the radiometer measurements then the actual profile will be different from the a-priori. This will result in errors which are higher than when the a-priori and estimated profiles are similar or the weather conditions for the two times are not very different. This difference in weather conditions is due to a weather phenomenon or a rain event.

Comment: Page 9, line 21-22: Describe what aspects of background information are correlated with measurements and why they are correlated? Response: This has been

added to Section 3.2.2 line 24 of page 7.

"Background information statistics here means the background information covariance information represented by the matrix S ̣_a and determines the range of variability information associated with water vapor or temperature profiles during a time period. Background data set is very important for the performance of the retrieval algorithm in terms of accuracy and ability to sense temporal changes. This is because background data set taken closer in time to the radiometer measurement will describe the atmospheric conditions i.e., the temperature and humidity profiles as well as the wind vector etc. during the measurements. So, the inversion of the measurements results in retrieval of the most persistent water vapor and temperature profile while being constrained by the background data set and a-priori. As a result, the retrieved profiles will detect similar features as the actual profile and hence will represent the gradients and dynamic changes in the actual profile. The atmospheric conditions during a particular season or month are correlated because the atmospheric conditions are similar throughout the time period accept a few outliers which cannot be correlated to the time of interest. Therefore, measurements along with the background data set and the a-priori will retrieve the most probable water vapor and temperature profile while an outlier might or might not be detected depending on whether that event is properly described by the background covariance matrix."

Please also note the supplement to this comment:
http://www.geosci-instrum-method-data-syst-discuss.net/gi-2016-16/gi-2016-16-AC5-supplement.pdf
* * *
[Figure]

**Figure 2. (a)** Weighting functions for measurement frequencies used for water vapor profile retrieval. **(b)** Weighting functions for measurement frequencies used for temperature profile retrieval.

**Fig. 1.** Figure 2

[Figure]

**Figure 5. Background information covariance matrix for 80 layers (100 m thick) (a)
water density (b) temperature profiles. The x and y axes are in kilometers for both the
figures.**

**Fig. 2.** Figure 5

Figure 3. The observation error covariance matrix for (a) water vapor frequency measurements (b) temperature measurements.

**Fig. 3.** Figure 3

[Figure]

[Figure]

**Figure 10. Retrieved profile sensitivity to observation error covariance matrix (a) Water vapor profile (b) Temperature profile**

**Fig. 4.** Figure 10

**Supplement:**

**Time Series Analysis of Ground-Based Microwave Measurements at K- and V-Bands to Detect Temporal Changes in Water Vapor and Temperature Profiles**

Sibananda Panda[1], Swaroop Sahoo[2], G. Pandithurai[3]

[1]School of Electronics Engineering, KIIT University, Odisha, India
[2]Department of Electrical Engineering, Indian Institute of Technology Palakkad, Palakkad, Kerala, India
[3]Indian Institute of Tropical Meteorology, Pune, India

*Correspondence to*: Swaroop Sahoo (swaroop.sahoo769@gmail.com)

**Abstract.** Ground-based microwave measurements performed at water vapor and oxygen absorption line frequencies are widely used for remote sensing of tropospheric water vapor density and temperature profiles, respectively. This paper focuses on using time series of independent frequency measurements at K- and V-bands along with statistically significant but short background data sets to retrieve and sense temporal variations and gradients in water vapor and temperature profiles. To study this capability, Indian Institute of Tropical Meteorology (IITM) had deployed a microwave radiometer at Mahabubnagar, Hyderabad during August 2011 as part of the Integrated Ground Campaign during the Cloud Aerosol Interaction and Precipitation Enhancement Experiment (CAIPEEX-IGOC). In this study, time series of water vapor and temperature profiles were retrieved using Bayesian optimal estimation method which uses Levenberg-Marquardt optimization technique. The temperature profiles for the first time have been estimated using optimized background information covariance matrix so as to improve the accuracy of the retrieved profiles as well as be able to detect gradients. Estimated water vapor and temperature profiles are compared with those taken from the reanalysis data updated by the Earth System Research Laboratory National Oceanic and Atmospheric Administration (NOAA), to determine the range of possible errors. Similarly, RMS errors are evaluated for the water vapor and temperature profiles for a month to estimate the accuracy of the retrievals. It is found that water vapor and temperature profiles can be estimated with an acceptable accuracy by using a background information data set compiled over a period of one month.

[revised manuscript text omitted]

**2.  INSTRUMENTS DEPLOYMENT**

Indian Institute of Tropical Meteorology (IITM) had deployed a microwave radiometer in Mahabubnagar (16° 44′ N, 77° 59′ E), Hyderabad for the whole month of August, 2011 as part of the Integrated Ground Campaign during the Cloud Aerosol Interaction and Precipitation Enhancement Experiment (CAIPEEX-IGOC) (Leena et. al., 2015). This is a frequency agile radiometer and operated at 8 frequencies in the range 22-30 GHz and 14 frequencies from 51.0 GHz to 58.0 GHz in V-band, at elevation angles of $15^o$, $90^o$ and $165^o$. The resolution of the instrument varies from 0.1 to 1 K depending on integration time i.e., 0.01 to 2.5 seconds (Radiometrics Corporation, 2008) . The accuracy of the brightness temperature measurements is approximately 0.2 K and the bandwidth of the channels is 300 MHz. This instrument also has a single channel infrared radiometer in addition to surface pressure, humidity and temperature sensors. The multichannel microwave radiometer is calibrated by injecting noise from a noise diode to remove the system gain fluctuations. Two sided tipping curve calibration method has been used to determine the brightness temperatures from the measured voltages for water vapor channels and the cold (liquid nitrogen) and hot load calibration (internal black body at ambient temperature) is used to calibrate the temperature channels measurements.

Radiometer measurements during the field campaign were performed throughout the day and night under varying atmospheric conditions which included clear and cloudy skies. The time series of calibrated brightness temperatures for 22.23, 25.0, 51.243 and 53.36 GHz are shown in Figure 1. It can be observed that brightness temperatures at 22.23 GHz are comparatively higher than those at 25 GHz. This is because 22.23 GHz is the water vapor resonance frequency and is more sensitive to water vapor in the atmosphere than 25 GHz which is significantly far away from the water vapor resonance frequency. Similarly, measurements at 53.36 GHz are higher than those at 51.243 GHz because of the proximity of 53.36 GHz to the oxygen complex. Thus, measurement frequencies are sensitive to water vapor and temperature to a varying extent. This can also be confirmed by analyzing the weighting functions corresponding to water vapor and temperature frequencies shown in Figure 2. Figure 2(a) shows that weighting function values for 22.234 GHz are higher than those at 25.00 GHz at altitudes above 2 km while weighting function values at 25 GHz have higher values than those at 22.234 GHz below 2 km. This is because the measurements at 22.234 GHz are comparatively more sensitive to changes in water vapor at altitudes above 2.5 km while those at 25.00 GHz are more sensitive to changes in water vapor below that altitude. However, the weighting function values at 22.234 GHz for altitude range 2.5-8 km are significantly higher than those at 25.00 GHz so that brightness temperatures at 22.234 GHz are still higher than those at 25.00 GHz.

The temperature measurement frequencies shown in Figure 2 (b) are most sensitive to temperature variations from 0 to 4 km altitude. 53.36 GHz weighting function (represented in green in Figure 2 b) is higher than 51.243 GHz weighting function (represented in blue in Figure 2 b) at all altitudes.

To complement the radiometer measurements, Vaisala RS92-SGP radiosondes were launched everyday at 12 UTC. These radiosondes were launched from the radiometer deployment location to provide vertical profiles (temporal resolution of two seconds) of relative humidity, temperature, dew point temperature, pressure, and wind. These radiosondes data have been used as the source of apriori information as well as the source of background dataset during this study and analysis.

**3. THEORETICAL BACKGROUND**

**3.1 Remote Sensing of Water Vapor and Temperature Profile**

Remote sensing of water vapor and temperature is based on the measurement of microwave radiation emitted by water vapor and oxygen molecules. The emission and absorption of microwave radiation due to water vapor and oxygen in each tropospheric layer change the microwave radiation that reaches the ground. This variation in radiation is due to the concentration of water vapor in the atmosphere and the temperature at various altitudes. Therefore, these microwave radiation reaching the ground are source of information about the humidity distribution and temperature variation in the atmosphere at different heights.

Measurement of this radiation at the weak water humdity absorption line (centred at 22.235 GHz) is used for the sensing of water vapor profile variation. This is based on humidity absorption line broadening. This broadening is due to motion of the water molecules and their collisions with other water molecules and is known as pressure broadening. Thus change in pressure has a significant impact on the width of the absorption lines as well as the absorption values. So, a decrease in the atmospheric pressure reduces the line width and increases the water vapor absorption line strength which is most prominent at 22.235 GHz (the center of the absorption line). Therefore, closer the proximity of the measurement frequency to the weak water vapor resonance frequency higher the sensitivity to water vapor at high altitudes. As the pressure increases the absorption line widens resulting in reduced sensitivity to water vapor at high altitudes. However, frequencies farther away from the center frequency are more sensitive to water vapor changes close to ground level. This is again proven by the weighting functions at various frequencies. Weighting functions closest to the water vapor resonance frequencies are almost twice more sensitive to water vapor at 8 km than near ground level. While frequencies further way from the resonance peak are most sensitive to changes close to ground level. Therefore, a combination of various frequency measurements is able to detect the profile information about water vapor.

Similarly, microwave radiation from oxygen at the 60 GHz absorption complex can be used for retrieving temperature profile information because atmospheric absorption in the 50-75 GHz range is primarily due to oxygen molecules. The absorption due to oxygen molecule is due to magnetic moment 33 spin-rotational lines between 51.5-67.9 GHz. These spin-rotational lines blend together at lower altitude due to the pressure broadening of the lines. This blended absorption lines has a shape similar to an absorption band centered at 60 GHz. However, the absorption line intensity is not the simple addition of

isolated line intensities but the "overlap interference" which gives rise to a very complex absorption band called the oxygen complex. As a result the opacity at the 60 GHz is significantly higher than that at 50 GHz, so the radiometer just observes the radiation emitted close to the ground surface. To sample the whole troposphere measurements need to be performed at a number of frequencies away from the center frequency.

5    Since, oxygen is the most well mixed gas in the atmosphere and its proportion in the atmosphere is almost constant and altitude independent from ground level to 80 km, the microwave radiation at the oxygen absorption lines contains atmospheric temperature profile information.

**3.2 Retrieval Techniques**

10    **3.2.1    Bayesian Optimal Estimation**

The Bayesian optimal estimation is an inversion method using multiple K- and V-band microwave frequency measurements to retrieve profiles of humidity and temperature. This is because various measurement frequencies have varying sensitivity to water vapor and temperature at various altitudes. These multiple measurements combined with the a-priori information and a background information covariance matrix can be used for retrieving water vapor and temperature profiles while sensing the

15    associated temporal variations and gradients.

The retrieval of water vapor and temperature profiles from brightness temperature measurements is a non-linear problem; a variation of the iterative Gauss-Newton method is required for the estimation process. This retrieval technique uses a-priori humidity and temperature information as a constraint to determine an unique solution to the inverse problem. A-priori in this paper represents the measurement of water vapor and temperature profiles prior to the radiometer brightness temperature

20    measurements. This is also known as the initialization profile in this paper. The a-priori information is taken from radiosonde launched a few hours before the radiometer performs the measurement.

In addition to the a-priori information, water vapor density and temperature background information statistics is also introduced as the inversion constraint. These statistics provide variability information associated with the atmospheric humidity and temperature profiles as well as the inter layer correlation during a particular time period. The number of

25    elements in the background data set and the relationships among them determines the values of the background information covariance matrix elements. Since, in this study the dataset used for calculating the background statistics has been taken close to measurement time it will be more representative of weather conditions during that time period and location. Background information statistics here means the background information covariance information represented by the matrix $\bar{\bar{S}}_a$.

30    The Bayesian optimal estimation uses the Levenberg-Marquardt (LM) optimization method (Rodgers, 2000). The LM technique is shown in Eqs. (1).

$$\bar{x}_{i+1} = \bar{x}_i + \left((1+\gamma)\bar{\bar{S}}_a^{-1} + \bar{\bar{K}}_i^T\bar{\bar{S}}_\epsilon^{-1}\bar{\bar{K}}_i\right)^{-1}\left(\bar{\bar{K}}_i^T\bar{\bar{S}}_\epsilon^{-1}[\bar{T}_B' - \bar{\bar{T}}_B(\bar{x}_i)] - \bar{\bar{S}}_a^{-1}[\bar{x}_i - \bar{x}_a]\right) \tag{1}$$

where i is the iteration index, $\overline{\overline{K}}_i$ is the kernel or weighting function matrix and determines the sensitivity of the measurements at various frequencies to changes in the parameter of interest at various altitudes, $\overline{x}_i$ is the water vapor density or temperature profile which is updated at each iteration and is same as initialization profile for i =1, $\overline{T}'_B$ is the measured brightness temperature vector at water vapor density or temperature measurement frequencies, $\overline{x}_a$ is the background profile and is same as the initialization profile in this case because a small dataset is used as background dataset. $\overline{T}_B(\overline{x}_i)$ is the radiative transfer model simulated brightness temperature using the absorption coefficients calculated from a Rosenkranz model (Rosenkranz, 1993) (Rosenkranz, 1998).

$\overline{\overline{S}}_\epsilon$ is the observation error covariance matrix and contains the uncertainty information associated with the measurement. The observation error covariance matrix takes into consideration the radiometric measurement noise, representativeness error and radiative transfer model errors. The diagonal elements of the observation error covariance matrix are approximately in the range of 0.23 to 0.29 $K^2$ and some of the off-diagonal elements are close to zero. The observation error covariance, R, determines the uncertainty associated with the observations. This uncertainty has contributions from radiometric noise (E), forward model (F) and representativeness (M) errors. Radiometric noise is determined based on radiometric resolution which is the minimum difference in scene brightness temperature that can be sensed by the receiver. This value for MP3000-A varies from 0.1 to 1 K depending on integration time (Radiometrics Corporation, 2008). The typical value is 0.25 K for each measurement frequency while considering an integration time of 250 msec. In addition to the radiometric noise, forward model errors introduced due to inadequate absorption models. These are determined using the difference between brightness temperatures simulated by two absorption models i.e., the Rosekranz model as well as MPM93 (Liebe et al., 1993) which are then used to determine the forward model error covariance matrix. Another source of uncertainty is the representativeness error which takes into consideration the radiometer's sensitivity to fluctuations in the atmosphere on a time scale shorter than that can be represented by any numerical weather prediction model or radiosondes profiles. The representative covariance is calculated in Eq. 2.

$$M = E(\overline{T}'_B(t + \Delta t) - \overline{T}'_B(t))(\overline{T}'_B(t + \Delta t) - \overline{T}'_B(t))^T \qquad (2)$$

where t is time and $\Delta t$ is the time scale of difference. The observation error covariance matrix is shown in Figure 3 and given by $R = E + F + M$.

$\overline{\overline{S}}_a$ is the background covariance matrix which is computed using information from 50 radiosonde profiles launched over a period of one month. $\gamma$ is the LM factor and the value of $\gamma$ is updated at each iteration based on value of $J(x)$ from Eq. (3). Various initial values of $\gamma$ in the range of $\gamma = 1$ and $\gamma = \infty$ have been considered for starting of the iteration. For $\gamma = 1$, the iteration might move towards a local minima while in case of $\gamma = \infty$ the iteration immediately moves towards the global minima which gives a solution which does not converge. Therefore, the initial value of $\gamma$ is assumed to be one. It was observed that the algorithm did not converge with a valid output for this initial value of gamma so the initial value of gamma is increased at regular intervals to check the convergence. It was found empirically, that gamma with an initial value of 5000 converges the algorithm for all cases. As part of the iteration if the value of $J(x)$ increases, then the iteration is discarded and

the value of γ is increased 10 fold and the iteration is repeated. This is done so as to discard any invalid output which could be close to one of the local minima. If value of J(x) decreases, then the iteration is valid and the value of γ is reduced by a factor of 2 for the next iteration even if the convergence criteria is not satisfied (Hewison, 2007). This process is followed until the convergence criterion is validated by the output profile. This process is illustrated in Figure 4. It can be observed

5 that as the cost function decreases, the gamma value decreases and vice versa. At local minima of the cost function the gamma value also reduces.

LM technique output is dependent on the cost function represented by J(x) in Eq. (3)

$$J(x) = [\bar{x} - \bar{x}^b]^T \bar{\bar{S}}_a^{-1} [\bar{x} - \bar{x}^b] + [\bar{T}_B(\bar{x}_i) - \bar{T}_B']^T \bar{\bar{S}}_\epsilon^{-1} [\bar{T}_B(\bar{x}_i) - \bar{T}_B'] \qquad (3)$$

where $\bar{x}^b$ and $\bar{x}$ are the initialization profiles (either water vapor or temperature) and output profiles (either water vapor or temperature) for each iteration, respectively. The final water vapor or temperature output profiles are determined by the

10 convergence criterion given by Eq. (4)

$$[\bar{T}_B(\bar{x}_{i+1}) - \bar{T}_B(\bar{x}_i)]^T \bar{\bar{S}}_{\delta y}^{-1} [\bar{T}_B(\bar{x}_{i+1}) - \bar{T}_B(\bar{x}_i)] \ll m \qquad (4)$$

where $m$ is 5 and 7 (dimension of water vapor and temperature measurement vector) for water vapor and temperature profile retrieval and $\bar{\bar{S}}_{\delta y}$ is the covariance between $\bar{T}_B'$ and $\bar{T}_B(\bar{x}_i)$. Eq. (4) determines the termination of the iterative process. The iteration stops when Eq. (4) reaches a value q which is very small in comparison to m. Therefore, the value of q is chosen to be 0.05 and 0.07 for water vapor and temperature profile retrieval, respectively, which is 1/100 times the number of

15 measurements used).

**3.2.2 Impact of Background Dataset on Retrieval**

As already studied and determined by Scheve (Scheve & Swift, 1999), Hewison(Hewison, 2007), Solheim (Solheim, et al., 1998) and Sahoo (Sahoo et. al., 2015a) the number of measurement frequencies which provide altitude related information

20 about water vapor and temperature are limited by the information content or degrees of freedom of the measurements. Thus, use of these measurements in various inversion methods to retrieve the thermodynamic properties at more number of altitudes than the information content limit is an ill-posed as well as a non-linear problem. To overcome these short comings Bayesian optimal estimation method uses background information statistics.

Background information statistics here means the background information covariance information represented by the matrix

25 $\bar{\bar{S}}_a$ and determines the range of variability information associated with water vapor or temperature profiles during a time period. Background data set is very important for the performance of the retrieval algorithm in terms of accuracy and ability to sense temporal changes. This is because background data set taken closer in time to the radiometer measurement will describe the atmospheric conditions i.e., the temperature and humidity profiles as well as the wind vector etc. during the measurements. So, the inversion of the measurements results in retrieval of the most persistent water vapor and temperature

30 profile while being constrained by the background data set and a-priori. As a result, the retrieved profiles will detect similar features as the actual profile and hence will represent the gradients and dynamic changes in the actual profile. The

atmospheric conditions during a particular season or month are correlated because the atmospheric conditions are similar throughout the time period accept a few outliers which cannot be correlated to the time of interest. Therefore, measurements along with the background data set and the a-priori will retrieve the most probable water vapor and temperature profile while an outlier might or might not be detected depending on whether that event is properly described by the background covariance matrix.

The background information covariance matrix used in this paper is shown in Figure 5 (a) and (b) and are calculated using water vapor density and temperature profiles measured over a period of one month, respectively. It can be observed that most of the water vapor variability information is between 20-40 layers which correspond to the altitude range of 2 to 4 km. On the other hand, Figure 5 (b) shows that the temperature variability information is primarily below an altitude of one km and also in the range of 2-4.2 km. In contrast to these results, when background information covariance matrix is computed from a large dataset, important weather events or temporally varying conditions are overshadowed because the covariance matrix takes into the consideration the overall variability information while reducing the weight of certain weather conditions which correspond to a particular season (Sahoo et.al., 2015b).

The goal of this study is to retrieve water vapor and temperature profiles with improved accuracy while using a background dataset measured over a period of one month so as to detect the temporal changes and gradients in the lowest 8 km of the profiles.

**3.3 Neural Network Estimation**

Estimation of water vapor and temperature profiles from microwave radiometer brightness temperatures is done using a proprietary neural network method (NN) developed by Radiometrics Corporation (Solheim, et al., 1998). NN zenith estimation of temperature, water vapor density, relative humidity, and liquid water content profiles are performed simultaneously from the radiometer measurements plus the infrared (IR) channel. The retrieved profiles are estimated at 58 height levels, with 50 meter steps from the surface up to 500 m, then 100 m steps to 2 km, and 250 m steps from 2 to 10 km. However, it has to be noted that above approximately 7 km, the atmospheric water vapor density and temperature approach the climatological mean values.

As part of the retrieval process the NN is trained using a back-propagation algorithm and radiosonde data which has been collected over a period of time i.e., usually 4 to 5 years using. The radiosonde data that is used for training the network is taken from one or more sites with the same climatological conditions as the observation site. The radiosonde profiles are used for simulating the brightness temperature using absorption models and radiative transfer equations. The NN estimation uses a standard feed-forward network (Radiometrics Corporation, 2008) to retrieve the temperature, humidity and liquid water profile that is most consistent with the atmospheric conditions and radiometric measurements.

[revised manuscript text omitted]
. In addition to that, it can be observed that some of the retrieved profiles in Figure 8 (b) showed higher than usual errors i.e., 2 $g/m^2$ and above. This is because the water vapor profile retrieval accuracy is significantly affected by the a-priori profile as shown by Sahoo et al., 2015. If the atmospheric conditions during the a-priori profile measurement (radiosonde launch) are very different from the conditions during the radiometer measurements then the actual profile will be different from the a-priori. This will result in errors which are higher than when the a-priori and estimated profiles are similar or the weather conditions for the two times are not very different. This difference in weather conditions is due to a weather phenomenon or a rain event.

The range of errors associated with the temperature profiles are shown in Figure 9 (a) and (b) for both the NN and Bayesian optimal techniques, respectively. The error associated with the neural network profile is in the range of -3 to 5 K in the lowest 1 km of the troposphere and then the range changes to -4 to -8 K at 4 km above ground level. It is clear that the NN zenith retrieval underestimates the value of the temperature profile. The error associated with the Bayesian optimal estimated profile is shown in Figure 9 (b) and is in the range of -1 to 0 K except in the case of a few profiles. As in the case of water vapor profile, the errors associated with temperature profiles by Bayesian optimal estimation are significantly less than NN estimated profiles. The Bayesian optimally estimated retrievals using radiometer observations compare well with the reanalysis data in the lowest 2 km of troposphere because of the retrieval being constrained by apriori and surface measurements provided by the radiometer.

In addition, another analysis was performed to determine the sensitivity of retrieved profile to changes in observation error covariance matrix in the Bayesian optimal estimation. Water vapor and temperature profiles have been already retrieved for various days as shown in Figure 6 and Figure 7. The profile retrieved for 7-August-2011 has been reanalyze where the profile is retrieved using the observation covariance matrix shown in Figure 3. Again the profile is retrieved after increasing the diagonal element of the observation error covariance matrix by 0.25 $K^2$. The retrieved profiles for water vapor and temperature profile for both the observation error covariance matrices are shown in Figure 10. It can be observed that the retrieved water vapor profile for the new covariance matrix has higher error than the profile for the covariance matrix shown in Figure 3. The increase in error for the retrieved water vapor profile is in the range of 0.3 to 1.9 $g/m^3$ (0 to 8 km altitude) as the diagonal elements are increased by 0.25 $K^2$. Similarly, the increase of 0.25 $K^2$ in the diagonal elements of the temperature observation covariance matrix shown in Figure 3 increases the temperature error by 0.2 to 0.5 K (0 to 8 km altitude). Thus, the observation error covariance matrix has a significant impact on the retrieved profile quality and accuracy. Another analysis was performed to determine the deviation of the retrieved profiles from the reanalysis profile, root mean squared (RMS) errors are calculated for both the methods. RMS errors are calculated by comparing radiometer retrieved humidity and temperature profiles (retrieved using both Bayesian optimal estimation and NN method) with the reanalysis data (which is used as truth in this case) and are shown in Figure 11. Figure 11 (a) shows that RMS error associated with Bayesian optimal estimated water vapor profile varies from 0.2 to 0.4 $g/m^3$ for the lowest 4 km of the troposphere and is less than 0.2 $g/m^3$ above 5 km altitude. On the other hand, RMS error for NN retrieved profile is in the range of 1-2.5 $g/m^3$ in the lowest 2 km and is less than 1 $g/m^3$ above 4 km. Thus, the RMS error for water vapor profile retrieved using Bayesian optimal estimation is better than NN.

It can be observed from Figure 11 (b) that the RMS error for Bayesian optimal estimated temperature profiles is less than 0.6 K at any altitude from 0 to 8 km above ground level. However, the NN zenith retrieval error range is 1-2 K for lowest 2 km and then increases consistently above 2 km. The maximum error is approximately 7.5 K at 8 km above ground level. Thus, the Bayesian retrieval algorithm performs significantly better than NN zenith also for estimating temperature profile.

It has to be observed that that the RMS error for NN estimated water vapour density profile has a decreasing behaviour with altitude whereas the temperature profile RMS error increases with the height for the temperature profile. This is because NN

algorithm used to retrieve the water vapor and temperature profiles has been trained using a data set which has been taken from areas which had similar weather conditions as the radiometer observation site. However, two sites at the same altitude and longitude may have significantly different weather depending on the general conformation of the mountains in the area, the marine currents as well as the advection processes. This could lead to biases in the training of the radiometer algorithm which in turn would increase the error of the retrieved profile. This is what is causing the retrieval errors for both the water vapor and temperature profiles at the lower altitudes. However, at high altitudes the range of water vapor density values which are possible are limited and close to zero as they cannot be less than zero, (obviously the climatological mean) due to which the error levels reduce at high altitudes as shown in Figure 11(a). This is not the case for temperature profiles which can have really low values at high altitudes based on training data. Therefore, the temperature profiles have a high level of bias, hence the increase in error as the altitude increases.

**5. CONCLUSION AND DISCUSSION**

This paper comprehensively describes the Bayesian optimal estimation and the improvements applied to the technique to estimate humidity and temperature profiles with increased accuracy. The Bayesian technique, is an optimal combination of ground-based observations and the related background information, hence it retains the information carried by the background dataset which provides the variability information in the lower troposphere. The background dataset is one of the important parameters in improving accuracy whose effect has been studied in great detail. The technique is applied to the radiometer measurements performed for the month of August 2011 to retrieve water vapor density and temperature profiles. The retrieved profiles show that gradients can be detected along with temporal changes. The retrieved profiles have been compared with those from the NN method and also with the NOAA reanalysis data which is considered as truth in this case. The results show that Bayesian optimal estimation using a small background information dataset (50 profiles taken over a period of one month) has better performance than the NN method (which requires a large background dataset taken over 4-5 years as training data) when a large background data set is not available to train the algorithm. This is because the profiles in the background dataset are temporally and spatially correlated with the measurements performed by the radiometer. Thus, the most persistent profile is retrieved and the Bayesian optimal estimation achieves the improved retrieval performances throughout the altitude of interest.

Water vapor profiles retrieved using the Bayesian optimal estimation technique (Figure 6) compares well with the reanalysis data for 16-August-2011 and 26-August-2011 with differences less than 1.5 $g/m^3$ for the whole profile and for other days the difference is lower than the error observed for NN from ground to 3 km altitude. For most of the days the absolute errors are less than 2 $g/m^3$. In addition to that retrieved profiles are able to detect the gradients in the water vapor profile which are smoothed by the reanalysis data. On the other hand the RMS error for Bayesian estimation is less than 0.8 $g/m^3$ from ground to 8 km altitude which is less than those observed for NN. So the water vapor profile can be retrieved with an accuracy of better than 1.5 $g/m^3$. Temperature profiles estimated using Bayesian optimal estimation have been observed in Figure 7 to have differences of less than 3 K when compared with the reanalysis data while the NN profiles usually have a difference of

3 K or more for the whole profile. However, on most of the days temperature profiles can be retrieved at an accuracy of better than 1.5 K while detecting the gradients. This has been again proved in the RMS error analysis in Figure 11. The RMS error shows that Bayesian method has error less than 0.7 K while the NN has error higher than 2 K and increases as the altitude increases.

5 Along with other analyses, one has been performed to determine the sensitivity of retrieved profile accuracy to change in observation error covariance matrix. It has been observed that water retrieved profile error increases by almost 1-2 $g/m^3$ with an increase of 0.25 $K^2$ of the diagonal elements of the matrix. However, an increase of 0.25 $K^2$ of the diagonal elements of the temperature error covariance matrix results in an increase of error less than 0.8 K.

10 **ACKNOWLEDGEMENT**

Authors would like to thank Earth System Research Laboratory, National Oceanic and Atmospheric Administration (NOAA), for providing such an useful reanalysis dataset which helped in analyzing the error associated with the estimated profiles. We would also like to thank Dr. Xavier Bosch-Lluis for his important contribution and suggestions.

**FIGURES**

[Figure]

(a)  (b)

(c)  (d)

**Figure 1.Time series of brightness temperature at 22.23, 25.0, 51.243 and 53.36 GHz.**

[Figure]

Figure 2. (a) Weighting functions for measurement frequencies used for water vapor profile retrieval. (b) Weighting functions for measurement frequencies used for temperature profile retrieval.

[Figure]

(a)                                                        (b)

Figure 3. The observation error covariance matrix for (a) water vapour frequency measurements (b) temperature measurements.

[Figure]

**Figure 4. The value of cost function and gamma with respect to number of iterations are shown in the top and bottom figure respectively.**

[Figure]

(a)                                                        (b)

5   **Figure 5. Background information covariance matrix for 80 layers (100 m thick) (a) water density (b) temperature profiles. The x and y axes are in kilometers for both the figures.**

[Figure]

[Figure]

**Figure 6. Time series analysis data of water vapor retrieved profiles.**

[Figure]

**Figure 7. Time series analysis data of temperature retrieved profiles.**

[Figure]

**Figure 8. Error associated with water vapor density profile retrieved by(a) neural network (b) Bayesian optimal estimation.**

[Figure]

**Figure 9. Error associated with temperature profilesretrieved by(a) neural network(b) Bayesian optimal estimation.**

[Figure]

**Figure 10. Retrieved profile sensitivity to observation error covariance matrix (a) Water vapor profile (b) Temperature profile**

[Figure]

**Figure 11: RMS Error analysis for (a) water vapor profiles and (b) temperature profiles.**